# Label-wise Aleatoric and Epistemic Uncertainty Quantification

**Yusuf Sale**[1,3]    **Paul Hofman**[1,3]    **Timo Löhr**[1,3]    **Lisa Wimmer**[2,3]    **Thomas Nagler**[2,3]    **Eyke Hüllermeier**[1,3]

[1]Institute of Informatics, LMU Munich, Munich, Germany
[2]Department of Statistics, LMU Munich, Munich, Germany
[3]Munich Center for Machine Learning

## Abstract

We present a novel approach to uncertainty quantification in classification tasks based on label-wise decomposition of uncertainty measures. This label-wise perspective allows uncertainty to be quantified at the individual class level, thereby improving cost-sensitive decision-making and helping understand the sources of uncertainty. Furthermore, it allows to define total, aleatoric, and epistemic uncertainty on the basis of non-categorical measures such as variance, going beyond common entropy-based measures. In particular, variance-based measures address some of the limitations associated with established methods that have recently been discussed in the literature. We show that our proposed measures adhere to a number of desirable properties. Through empirical evaluation on a variety of benchmark data sets – including applications in the medical domain where accurate uncertainty quantification is crucial – we establish the effectiveness of label-wise uncertainty quantification.

## 1   INTRODUCTION

Thanks to methods of unprecedented predictive power, machine learning (ML) is becoming more and more ingrained into peoples' lives. It increasingly supports human decision-making processes in fields ranging from healthcare [Lambrou et al., 2010, Senge et al., 2014, Yang et al., 2009, Mobiny et al., 2021] and autonomous driving [Michelmore et al., 2018] to socio-technical systems [Varshney, 2016, Varshney and Alemzadeh, 2017]. The safety requirements of such applications trigger an urgent need to report *uncertainty* alongside model predictions [Hüllermeier and Waegeman, 2021]. Meaningful uncertainty estimates are indispensable for trust in ML-assisted decisions as they signal when a prediction is not confident enough to be relied upon.

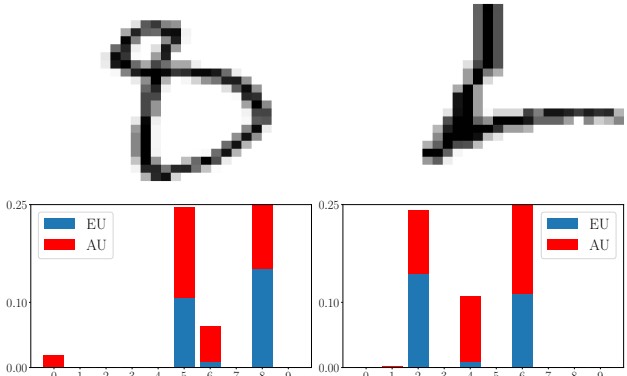

Figure 1: Label-wise *aleatoric* and *epistemic* uncertainties for MNIST instances.

In order to address predictive uncertainty about a query instance $x$ (e.g., an image like in Fig. 1), it is often crucial to identify its source. For one, uncertainty can arise through inherent stochasticity of the data-generating process, omitted variables or measurement errors [Gruber et al., 2023]. As such, *aleatoric* uncertainty (AU) is a fixed but unknown quantity. In addition, a lack of knowledge about the best way to model the data-generating process induces *epistemic* uncertainty (EU). Under the assumption that the model class is correctly specified, collecting enough information will reduce the EU until it vanishes in the limit of infinite data [Hüllermeier and Waegeman, 2021]. The attribution of uncertainty to its sources can inform decisions in various ways. For instance, it might help practitioners realize that gathering more data will be futile when only AU is present, or guide sequential learning processes like active learning [Shelmanov et al., 2021, Nguyen et al., 2022] and Bayesian optimization [Hoffer et al., 2023, Stanton et al., 2023] by seeking out promising parts of the search space that can be explored to reduce EU while avoiding uninformative areas with high AU.

Quantifying both AU and EU necessitates a meaningful uncertainty *representation*. In supervised learning, we consider

hypotheses in the form of probabilistic classifiers $h$ that map a query instance $\boldsymbol{x}$ to a probability distribution $p = h(\boldsymbol{x})$ on the label space. This prediction provides an estimate of the ground-truth (conditional) probability $p^*(\cdot \mid \boldsymbol{x})$, i.e., $p(y)$ estimates the true probability $p^*(y \mid \boldsymbol{x})$ of observing class label $y$ as outcome given $\boldsymbol{x}$. When predicting a single class label in a deterministic way, $p$ will be a Dirac measure. The case of probabilistic classification, which we study in this work, is more informative in the sense that a (posterior) probability is associated with all possible class labels, giving rise to a natural notion of AU around the observed outcome $y$. However, such probabilistic expressions are point predictions (in the space of probability distributions) derived from a single hypothesis $h$ learned on the training data. Since all other candidates in the hypothesis space are discarded in the process, $p$ cannot, by design, represent EU [Hüllermeier and Waegeman, 2021].

Expressing EU requires a further level of uncertainty representation. A straightforward approach is to impose a *second-order* distribution, effectively assigning a probability (density) to each candidate first-order distribution $p$, and equate the dispersion of this distribution with EU. Both the classical Bayesian paradigm [Gelman et al., 2013] and evidential deep learning (EDL) methods [Ulmer et al., 2023] follow this idea. As an alternative, methods founded on more general theories of probability, such as imprecise probabilities [Walley, 1991, Augustin et al., 2014], have been considered [Corani et al., 2012, Sale et al., 2023b].

In recent years, probabilistic classification has increasingly embraced a bi-level distributional approach, with a predominant reliance on *Shannon entropy* to dissect uncertainty into its aleatoric and epistemic components [e.g., Kendall and Gal, 2017, Smith and Gal, 2018, Charpentier et al., 2022]. In this approach, the entropy of the categorical output distribution over class labels is associated with the total predictive uncertainty for a query instance $\boldsymbol{x}$. By a well-known result from information theory [Cover and Thomas, 1999], this quantity decomposes additively into *conditional entropy* (representing AU) and *mutual information* (representing EU). While this set of measures may seem concise and intuitive, Wimmer et al. [2023] recently pointed out that it does not fulfill certain properties that one would naturally expect to hold. Whereas these methods solely focus on quantifying uncertainty at a *global* level, we argue that this perspective may not suffice for all decision-making scenarios. To address this gap, we propose an approach centered on *label-wise* uncertainty quantification. This approach allows for a more nuanced understanding of uncertainty, enabling decision-makers to evaluate the uncertainty associated with individual class predictions.

By adopting a label-wise perspective, our method facilitates more informed decision-making, especially in contexts where the consequences of incorrect predictions—as in medical scenarios—differ between classes.

This perspective on uncertainty quantification not only preserves the global perspective inherent in traditional approaches, but also enhances it by providing insights at the class level. Moreover, since a label-wise decomposition of uncertainty measures effectively amounts to reducing multinomial to binary classification, our approach is no longer restricted to uncertainty measures for categorical variables, such as entropy. Instead, it is amenable to a much broader class of measures, including variance as arguably the most common statistical measure of dispersion.

Our contributions are as follows:

**(1)** We propose a *label-wise* perspective enabling reasoning about uncertainty at the individual class level, aiding decision-making especially in scenarios where the stakes of incorrect predictions vary across classes. To this end, we leverage entropy- and variance-based measures for label-wise uncertainty quantification.

**(2)** We showcase that adopting a label-wise perspective retains the *global* perspective at the same time. In this regard, we demonstrate that the proposed measures satisfy a set of desirable properties, enhancing their theoretical appeal. In particular we show that our proposed variance-based measures overcome the drawbacks of the entropy-based approach, recently highlighted in the literature, without sacrificing practical applicability.

**(3)** Through empirical evaluation, we validate the efficacy of our approach, demonstrating its competitiveness in (global) downstream tasks such as prediction with abstention and out-of-distribution detection. Our empirical findings are substantiated across a range of classical machine learning benchmarks and verified in the medical domain, where suitable uncertainty quantification is indispensable.

Proofs of our theoretical results can be found in Appendix A. For experimental details and supplementary experiments, refer to Appendix B and Appendix C, respectively.

## 2 QUANTIFYING SECOND-ORDER UNCERTAINTY

In the following, we will be concerned with the supervised classification scenario. We refer to $\mathcal{X}$ as *instance* space, and we assume categorical target variables from a finite *label* space $\mathcal{Y} = \{y_1, \ldots, y_K\}$, where $K \in \mathbb{N}_{\geq 2}$. Thus, each instance $\boldsymbol{x} \in \mathcal{X}$ is associated with a conditional distribution on the measurable space $(\mathcal{Y}, 2^{\mathcal{Y}})$, such that $\theta_k := p(y_k \mid \boldsymbol{x})$ is the probability to observe label $y_k \in \mathcal{Y}$ given $\boldsymbol{x} \in \mathcal{X}$. Further, we note that the set of all probability measures on $(\mathcal{Y}, 2^{\mathcal{Y}})$ can be identified with the $(K-1)$-simplex $\Delta_K := \{\boldsymbol{\theta} = (\theta_1, \ldots, \theta_K) \in [0,1]^K \mid \|\boldsymbol{\theta}\|_1 = 1\}$. Consequently, for each $\boldsymbol{\theta} \in \Delta_K$, an associated degree of aleatoric uncertainty can be calculated.

To effectively represent epistemic uncertainty, it is necessary for the learner to express its uncertainty regarding $\boldsymbol{\theta}$. This can be achieved by a second-order probability distribution over the first-order distributions $\boldsymbol{\theta}$.

Two popular methods to obtain a second-order (predictive) distribution are Bayesian inference and Evidential Deep Learning. In both approaches, we arrive at a second-order predictor $h_2 : \mathcal{X} \longrightarrow \Delta_K^{(2)}$, where $\Delta_K^{(2)}$ denotes the set of all probability measures on $(\Delta_K, \sigma(\Delta_K))$; we call $Q \in \Delta_K^{(2)}$ a second-order distribution. For the sake of simplicity, we omit the conditioning on the query instance $\boldsymbol{x}$ in the notation. Hence, given an instance $\boldsymbol{x}$, the second-order distribution $Q$ represents our probabilistic knowledge about $\boldsymbol{\theta}$, i.e., $Q(\boldsymbol{\theta})$ is the probability (density) of $\boldsymbol{\theta} \in \Delta_K$. In the remainder of this paper, we assume that a second-order distribution $Q$ is already provided.

Given an uncertainty representation in terms of a second-order distribution $Q \in \Delta_K^{(2)}$, the subsequent question is how to suitably quantify total, aleatoric, and epistemic uncertainty. Popular approaches to uncertainty quantification in the literature [Houlsby et al., 2011, Gal, 2016, Depeweg et al., 2018, Smith and Gal, 2018, Mobiny et al., 2021] rely on information-theoretic measures derived from Shannon entropy [Shannon, 1948]. In the following section, we will revisit these commonly accepted entropy-based uncertainty measures and discuss meaningful properties that any uncertainty measure should possess.

## 2.1 ENTROPY-BASED MEASURES

We begin by revisiting the arguably most common entropy-based approach in machine learning for quantifying predictive uncertainty represented by a second-order distribution $Q$. This approach leverages (Shannon) entropy and its connection to mutual information and conditional entropy to quantify total, aleatoric, and epistemic uncertainty associated with $Q$.

Shannon entropy for a (first-order) probability distribution $\boldsymbol{\theta} \in \Delta_K$ is given by

$$\mathrm{H}(\boldsymbol{\theta}) := - \sum_{k=1}^{K} \theta_k \log_2 \theta_k \,. \tag{1}$$

Now, let $Y : \Omega \longrightarrow \mathcal{Y}$ be a (discrete) random variable, and denote by $\boldsymbol{\theta}_Y$ its corresponding distribution on the measurable space $(\mathcal{Y}, 2^{\mathcal{Y}})$. Then, we can analogously define the entropy of the random variable $Y$ by simply replacing $\theta_k$ in (1) by the respective distribution of $Y$. Entropy has established itself as an accepted uncertainty measure due to both appealing theoretical properties and the intuitive interpretation as a measure of uncertainty. In particular, it measures the uniformity degree of the distribution of a random variable.

Subsequently, following the notation of Wimmer et al. [2023], we assume that $\Theta \sim Q$. Therefore, $\Theta : \Omega \longrightarrow \Delta_K$ is a random first-order distribution which is distributed according to a second-order distribution $Q$, and consequently takes values $\Theta(\omega) = \boldsymbol{\theta}$ in the $(K-1)$-simplex $\Delta_K$.

Given a second-order distribution $Q$, we can consider its expectation given by

$$\bar{\boldsymbol{\theta}} := \mathbb{E}_Q[\Theta] = \int_{\Delta_K} \boldsymbol{\theta} \, \mathrm{d}Q(\boldsymbol{\theta}) \,, \tag{2}$$

which yields a probability distribution $\bar{\boldsymbol{\theta}}$ on $(\mathcal{Y}, 2^{\mathcal{Y}})$. This measure corresponds to the distribution of $Y$ when we view it as generated from first sampling $\Theta \sim Q$ and then $Y$ according to $\Theta$. Then, it is natural to define the measure of total uncertainty (TU) as the entropy (1) of $\bar{\boldsymbol{\theta}}$:

$$\mathrm{TU}(Q) := \mathrm{H}\left(\mathbb{E}_Q[\Theta]\right) \,. \tag{3}$$

Similarly, aleatoric uncertainty (AU) can be defined in terms of *conditional entropy* $\mathrm{H}(Y|\Theta)$:

$$\mathrm{AU}(Q) := \mathbb{E}_Q[\mathrm{H}(Y|\Theta)] = \int_{\Delta_K} \mathrm{H}(\boldsymbol{\theta}) \, \mathrm{d}Q(\boldsymbol{\theta}) \,. \tag{4}$$

By fixing a first-order distribution $\boldsymbol{\theta} \in \Delta_K$, all EU is essentially removed and only AU remains. However, as $\boldsymbol{\theta}$ is not precisely known, we take the expectation with respect to the second-order distribution. The measure of epistemic uncertainty is particularly inspired by the widely known additive decomposition of entropy into *conditional entropy* and *mutual information* (see also Section 2.4 in Cover and Thomas [1999]). This is expressed as follows:

$$\underbrace{\mathrm{H}(Y)}_{\text{entropy}} = \underbrace{\mathrm{H}(Y \,|\, \Theta)}_{\text{conditional entropy}} + \underbrace{\mathrm{I}(Y, \Theta)}_{\text{mutual information}}. \tag{5}$$

Rearranging (5) for mutual information yields a measure of epistemic uncertainty

$$\mathrm{EU}(Q) := \mathrm{I}(Y, \Theta) = \mathrm{H}(Y) - \mathrm{H}(Y \,|\, \Theta). \tag{6}$$

While entropy, mutual information, and conditional entropy provide meaningful interpretations for quantifying uncertainties within first-order predictive distributions, the suitability of these entropy-based measures for second-order quantification has been challenged by Wimmer et al. [2023]. This criticism was substantiated on the basis of a set of desirable properties, which will be discussed next.

## 2.2 DESIRABLE PROPERTIES

In this section we discuss desirable properties that any suitable uncertainty measure should fulfill. In the (uncertainty) literature it is standard practice to establish measures based on a set of axioms [Pal et al., 1993, Bronevich and Klir,

2008]. Such an axiomatic approach was also adopted in the recent machine learning literature [Hüllermeier et al., 2022, Sale et al., 2023a]. To this end, we revisit the axioms outlined by Wimmer et al. [2023], while also taking into account recently proposed properties that further refine the understanding of what constitutes a suitable measure of second-order uncertainty [Sale et al., 2023a]. Before discussing the proposed axioms, we first provide some mathematical preliminaries.

**Definition 2.1.** *Let $\Theta \sim Q$, $\Theta' \sim Q'$ be two random vectors, where $Q, Q' \in \Delta_K^{(2)}$. Denote by $\sigma(\Theta)$ the $\sigma$-algebra generated by the random vector $\Theta$. Then we call $Q'$*

(i) *a mean-preserving spread of $Q$, iff $\Theta' \stackrel{d}{=} \Theta + Z$, for some random vector $Z$ with $\mathbb{E}[Z \,|\, \sigma(\Theta)] = 0$ almost surely (a.s.) and $\max_k Var(Z_k) > 0$;*

(ii) *a spread-preserving location shift of $Q$, iff $\Theta' \stackrel{d}{=} \Theta + z$, where $z \neq 0$ is a constant;*

(iii) *a spread-preserving center-shift of $Q$, iff it is a spread-preserving location shift with $\mathbb{E}[\Theta'] = \lambda \mathbb{E}[\Theta] + (1 - \lambda)(1/K, \ldots, 1/K)^\top$ for some $\lambda \in (0, 1)$.*

*Note that for definitions (ii) and (iii) it should be guaranteed that the shifted probability measure $Q'$ remains valid within its support.*

Now, let TU, AU, and EU denote, respectively, measures $\Delta_K^{(2)} \to \mathbb{R}_{\geq 0}$ of total, aleatoric, and epistemic uncertainty associated with a second-order uncertainty representation $Q \in \Delta_K^{(2)}$. Wimmer et al. [2023] propose that any uncertainty measure should fulfill (at least) the following set of axioms:

A0  TU, AU, and EU are non-negative.

A1  $EU(Q) = 0$, if and only if $Q = \delta_{\boldsymbol{\theta}}$, where $\delta_{\boldsymbol{\theta}}$ denotes the Dirac measure on some $\boldsymbol{\theta} \in \Delta_K$.

A2  EU and TU are maximal for $Q$ being the uniform distribution on $\Delta_K$.

A3  If $Q'$ is a mean-preserving spread of $Q$, then $EU(Q') \geq EU(Q)$ (weak version) or $EU(Q') > EU(Q)$ (strict version), the same holds for TU.

A4  If $Q'$ is a spread-preserving center-shift of $Q$, then $AU(Q') \geq AU(Q)$ (weak version) or $AU(Q') > AU(Q)$ (strict version), the same holds for TU.

A5  If $Q'$ is a spread-preserving location shift of $Q$, then $EU(Q') = EU(Q)$.

Axiom A0 is an obvious requirement, ensuring that such measures reflect a degree of uncertainty without implying the absence of information or negative uncertainty, which would be conceptually unsound. Axiom A1 addresses the behavior of EU in the context of Dirac measures, where a Dirac measure $\delta_{\boldsymbol{\theta}}$ represents a scenario of complete certainty about $\boldsymbol{\theta} \in \Delta_K$. The vanishing of EU in this context aligns with the intuitive understanding that epistemic uncertainty should be zero when there is absolute certainty about the true underlying model. Further, Axiom A2 considers the condition under which EU and TU attain their maximal values, specifically when $Q$ is the uniform distribution on $\Delta_K$. This reflects situations of maximum uncertainty or ignorance, where the lack of knowledge about any specific outcome $\boldsymbol{\theta} \in \Delta_K$ leads to the highest level of uncertainty. As we will discuss later, this axiom is not without controversy, particularly in the fields of statistics and decision theory. Axiom A3 encapsulates the idea that spreading a distribution while preserving its mean should not reduce, and might increase, the epistemic (and thus, total) uncertainty. It underscores the notion that increased dispersion (while maintaining the mean) is associated with higher uncertainty, a concept that is central in statistics. Conversely, leaving the dispersion constant but shifting the distribution closer to the barycenter of the simplex, thereby expressing a belief about $\boldsymbol{\theta}$ that is closer to uniform, should be reflected by an increase in AU (Axiom A4). Lastly, Axiom A5 asserts that a spread-preserving location shift, which alters the distribution's location without affecting its spread, should leave the epistemic uncertainty unchanged. This property highlights the distinct nature of epistemic uncertainty, which is sensitive to the spread of the distribution rather than its location [Hüllermeier et al., 2022].

Taking into consideration recently proposed criteria for measures of second-order uncertainty [Sale et al., 2023a], we expand the existing set of Axioms A0–A5 by introducing two additional properties. For the set of all mixtures of second-order Dirac measures on first-order Dirac measures we write

$$\Delta_{\delta_m} = \left\{ \delta_m \in \Delta_K^{(2)} \,:\, \delta_m = \sum_{y \in \mathcal{Y}} \lambda_y \cdot \delta_{\delta_y}, \sum_{y \in \mathcal{Y}} \lambda_y = 1 \right\},$$

where $\delta_{\delta_y}$ denotes the second-order Dirac measure on $\delta_y \in \Delta_K$ for $y \in \mathcal{Y}$. Each element in this set should arguably have no aleatoric uncertainty, such that we postulate the following Axiom A6.

A6  $AU(\delta_m) = 0$ holds for any $\delta_m \in \Delta_{\delta_m}$.

Now, let $\mathcal{Y}_1$ and $\mathcal{Y}_2$ be partitions of $\mathcal{Y}$ and $Q \in \Delta_K^{(2)}$; further denote by $Q_{|\mathcal{Y}_i}$ the corresponding marginalized distribution for $i \in \{1, 2\}$.

A7  $TU_{\mathcal{Y}}(Q) \leq TU_{\mathcal{Y}_1}(Q_{|\mathcal{Y}_1}) + TU_{\mathcal{Y}_2}(Q_{|\mathcal{Y}_2})$, and the same holds for AU and EU.

Axiom A7 guarantees that the total uncertainty of a second-order distribution is bounded by the sum of total uncertainties of its corresponding marginalizations.

# 3 LABEL-WISE UNCERTAINTY QUANTIFICATION

In this section, we propose to measure uncertainty in a label-wise manner, and to obtain the overall uncertainty associated with a prediction by aggregating the uncertainties across the individual labels. This approach allows us to adopt a *label-wise* perspective while retaining the *global* one.

Let us emphasize again that the label-wise perspective is particularly useful in settings where decisions following the prediction of different labels are associated with unequal costs. For instance, when predicting the sub-type of a certain medical condition, with costly treatment administered at occurrence of one of the sub-types, the marginal uncertainty about this category might be of particular interest. We present some experimental results on medical images in Section 4.1. The global view, on the other side, is crucial in scenarios where understanding the overall uncertainty is key to making informed decisions. For instance, TU serves as an indicator for the overall reliability of the model for the given observation. Meanwhile, AU and EU distinguish between the uncertainty arising from the data's inherent variability and that stemming from the model's knowledge limitations, respectively.

We denote by $\boldsymbol{Y} : \Omega \longrightarrow \{0,1\}^K$ the $K$-dimensional random vector indicating the presence or absence of a particular label $y_k \in \mathcal{Y}$ for $k \in \{1, \ldots, K\}$. Further, define $\Theta_k := P(Y_k = 1)$ and assume that the random vector $\boldsymbol{\Theta} = (\Theta_1, \ldots, \Theta_K)$ is distributed according to a second-order distribution $Q \in \Delta_K^{(2)}$, i.e., $\boldsymbol{\Theta} \sim Q$. Moreover, let $Q_k$ be the marginal distribution of the random variable $\Theta_k$, such that $\Theta_k \sim Q_k$, and denote its expectation by $\bar{\theta}_k = \mathbb{E}[\Theta_k]$.

Our general approach to label-wise uncertainty quantification adheres to the following template: First, we define *local* measures of total, aleatoric, and epistemic uncertainty per label: $\mathrm{TU}(Q_k)$, $\mathrm{AU}(Q_k)$, $\mathrm{EU}(Q_k)$ for $k \in \{1, \ldots, K\}$. One way to define these measures in a meaningful way is to adopt a loss-based perspective: Consider a learner making probabilistic predictions $\hat{\theta}$ for a (binary) outcome $Y_k$, which are penalized with a loss $\phi(\hat{\theta}, Y_k)$. If $Y_k$ is distributed according to $\theta_k$, then the expected loss is given by

$$\phi(\hat{\theta}, \theta_k) := \mathbb{E}_{Y_k \sim \theta_k} \phi(\hat{\theta}, Y_k). \tag{7}$$

In our case, $\theta_k$ itself is presumably distributed according to the second-order distribution $Q_k$, so the prediction $\hat{\theta}$ induces the expected loss

$$\mathbb{E}_{\theta_k \sim Q_k} \phi(\hat{\theta}, \theta_k) = \mathbb{E}_{\theta_k \sim Q_k} \mathbb{E}_{Y_k \sim \theta_k} \phi(\hat{\theta}, Y_k). \tag{8}$$

Broadly speaking, the idea is as follows: If the expected loss (8) can be kept small, by virtue of an appropriate prediction $\hat{\theta}$, then this signifies a situation of low (total) uncertainty. Otherwise, if this is not possible, then the uncertainty is high. More specifically, we suggest the following definitions for the three types of uncertainty:

- Total uncertainty is the minimum of (8), i.e., the expected loss of the risk-minimizing prediction $\hat{\theta}$ given knowledge of the second-order distribution $Q_k$:

$$\mathrm{TU}(Q_k) := \min_{\hat{\theta}} \mathbb{E}_{\theta_k \sim Q_k} \phi(\hat{\theta}, \theta_k) \tag{9}$$

- Aleatoric uncertainty is the expected loss of the risk-minimizing prediction $\hat{\theta}$ given knowledge about the true $\theta_k$ (sampled from $Q_k$).

$$\mathrm{AU}(Q_k) := \mathbb{E}_{\theta_k \sim Q_k} \min_{\hat{\theta}} \phi(\hat{\theta}, \theta_k) \tag{10}$$

- Epistemic uncertainty is the difference between these two, i.e., the extra loss that is caused by the lack of knowledge about the true $\theta_k$:

$$\mathrm{EU}(Q_k) := \mathrm{TU}(Q_k) - \mathrm{AU}(Q_k) \tag{11}$$

In particular, total uncertainty reflects an optimistic perspective inherent in the idea of quantifying uncertainty in terms of *unavoidable loss*. To illustrate, let us consider the following: Given a second-order distribution $Q_k$, from which a distribution $\theta_k$ will be sampled, one aims to predict $\hat{\theta}$ and will then incur the loss $\phi(\hat{\theta}, \theta_k)$. The objective is to minimize the expected loss, hence the minimum in (9). Success in minimizing this expected loss implies that $Q_k$ is "peaked" or close to a Dirac measure, indicating low uncertainty. Conversely, if $Q_k$ is widely spread and not very informative, the uncertainty is high, and even the optimal prediction $\hat{\theta}$ cannot ensure a low loss. This explains the rationale behind total uncertainty (9).

The additive relationship of the (global) entropy-based measures has been a subject of debate in the literature [Wimmer et al., 2023]. In our framework, it can be justified as follows: As discussed before, TU represents the unavoidable loss in predicting $\theta_k$, incorporating an epistemic component since the true data-generating process $\theta_k$ is unknown and only characterized by $Q_k$. This epistemic uncertainty would vanish if $\theta_k$ were known, leaving only aleatoric uncertainty. With $\theta_k$ known, the best prediction aligns with $\hat{\theta} = \theta_k$, resulting in a loss $\phi(\theta_k, \theta_k)$; for instance, in the case of log-loss, this equates to Shannon entropy. Therefore, AU is defined as the expectation with respect to $Q_k$ of this residual loss, as per Equation (10). Consequently, EU is measured by the difference between TU and AU, indicating the extent (in expectation) to which the unavoidable loss can be mitigated by eliminating epistemic uncertainty.

Nevertheless, certain scenarios call for a *global* perspective on predictive uncertainty. To obtain corresponding measures, the most obvious idea is to define total, aleatoric, and epistemic uncertainty associated with a second-order distribution $Q \in \Delta_K^{(2)}$ by summing over all label-wise un-

certainties:

$$\mathrm{TU}(Q) := \sum_{k=1}^{K} \mathrm{TU}(Q_k) \tag{12}$$

$$\mathrm{AU}(Q) := \sum_{k=1}^{K} \mathrm{AU}(Q_k) \tag{13}$$

$$\mathrm{EU}(Q) := \sum_{k=1}^{K} \mathrm{EU}(Q_k) \tag{14}$$

As mentioned, one advantage of the label-wise decomposition, which goes hand in hand with a binarization of the problem (the $Y_k$ are binary outcomes), is that it broadens the scope of measures that can be applied. Our approach as outlined above is "parameterized" by the loss function $\phi$. Natural candidates for this loss are (strictly) proper scoring rules [Gneiting and Raftery, 2007], which have the meaningful property that the risk-minimizer $\hat{\theta}$ in (7) coincides with $\theta_k$ itself; therefore, total and aleatoric uncertainty become, respectively,

$$\mathrm{TU}(Q_k) = \phi(\bar{\theta}_k, \bar{\theta}_k) = \phi(\mathbb{E}[\Theta_k], \mathbb{E}[\Theta_k]) \tag{15}$$

$$\mathrm{AU}(Q_k) = \mathbb{E}_{\theta_k \sim Q_k} \, \phi(\theta_k, \theta_k) \tag{16}$$

Our construction extends the well-established information-theoretic decomposition of Shannon entropy into conditional entropy and mutual information, which are the most widely used measures for uncertainty quantification in machine learning. Specifically, when the loss function $\phi$ is the log-loss, entropy-based measures are recovered as a special case. Our generalization allows for the use of losses other than log-loss, such as variance, thereby broadening the scope and applicability of our framework. Furthermore, we demonstrate desirable properties of this generalization.

In the following, we propose two concrete instantiations: The log-loss $\phi(\hat{\theta}, Y) = -(Y \log(\hat{\theta}) + (1 - Y) \log(1 - \hat{\theta}))$, which leads to entropy as (total) uncertainty, and the squared-error loss $\phi(\hat{\theta}, Y) = (\hat{\theta} - Y)^2$, which leads to variance as uncertainty measure.

We note that variance is an example of a measure that can be applied to the binary case but not to the categorical case in general. However, our label-wise approach addresses this issue, enabling the effective use of variance-based uncertainty measures for classification purposes.

## 3.1 ENTROPY-BASED MEASURES

In complete analogy to global entropy-based measures (3), (4), and (5) we can define the corresponding label-wise counterparts for all $k \in \{1, \ldots, K\}$:

- Label-wise total uncertainty (15) is given by $\mathrm{H}(\bar{\theta}_k) = \mathrm{H}(\mathbb{E}[\Theta_k])$.

- Label-wise aleatoric uncertainty (16) is given by expected conditional entropy $\mathbb{E}[\mathrm{H}(Y_k \,|\, \Theta_k)]$.

- Label-wise epistemic uncertainty is given by the expected KL-divergence $\mathbb{E}[\mathrm{D}_{\mathrm{KL}}(\Theta_k \,||\, \bar{\theta}_k)]$.

The corresponding global measures (12–14) are then given as follows:

$$\mathrm{TU}(Q) := \sum_{k=1}^{K} \mathrm{H}(\mathbb{E}[\Theta_k]) \tag{17}$$

$$\mathrm{AU}(Q) := \sum_{k=1}^{K} \mathbb{E}[\mathrm{H}(Y_k \,|\, \Theta_k)] \tag{18}$$

$$\mathrm{EU}(Q) := \sum_{k=1}^{K} \mathbb{E}[\mathrm{D}_{\mathrm{KL}}(\Theta_k \,||\, \bar{\theta}_k)] \tag{19}$$

In the following we demonstrate which of the properties discussed in 2.2 are fulfilled by the entropy-based measures constructed in a label-wise manner.

**Theorem 3.1.** *Entropy-based measures* (17), (18), *and* (19) *satisfy Axioms A0, A1, A2 (only for TU), A3 (strict version), A4 (strict version, only for TU), A6, and A7.*

## 3.2 VARIANCE-BASED MEASURES

Here, we leverage the *law of total variance*: for any random variable $X \in L^2(\Omega, \mathcal{A}, P)$ and sub-$\sigma$-algebra $\mathcal{F} \subseteq \mathcal{A}$,

$$\mathrm{Var}(X) = \mathbb{E}[\underbrace{\mathbb{E}[(X - \mathbb{E}[X \,|\, \mathcal{F}])^2 \,|\, \mathcal{F}]}_{=: \mathrm{Var}(X \,|\, \mathcal{F})}] + \mathrm{Var}(\mathbb{E}[X \,|\, \mathcal{F}]) .$$

Then, observing that $\sigma(\Theta_k) \subseteq \mathcal{F}$ for any $k \in \{1, \ldots, K\}$, we get

$$\mathrm{Var}(Y_k) = \mathbb{E}[\mathrm{Var}(Y_k \,|\, \sigma(\Theta_k))] + \mathrm{Var}(\mathbb{E}[Y_k \,|\, \sigma(\Theta_k)])$$

$$= \mathbb{E}[\Theta_k \cdot (1 - \Theta_k)] + \mathrm{Var}(\Theta_k) .$$

This equality suggests an alternative definition of total uncertainty and its (additive) decomposition into an aleatoric and an epistemic part:

- Label-wise total uncertainty is given by $\mathrm{Var}(Y_k)$ and is obtained as an instantiation of (15) with $\phi$ the squared-error loss.

- Label-wise aleatoric uncertainty (16) is captured by $\mathbb{E}[\Theta_k \cdot (1 - \Theta_k)]$, reflecting the inherent randomness in the outcome of each $Y_k$. Just like conditional entropy, it can be seen as the (expected) "conditional variance" of $Y_k$ and corresponds to the expected squared-error loss provided the true value of $\Theta_k$ is given.

- Label-wise epistemic uncertainty is quantified by $\mathrm{Var}(\Theta_k)$. Just like mutual information corresponds to the expected reduction in log-loss achieved by the knowledge of $\Theta_k$, $\mathrm{Var}(\Theta_k)$ is the expected reduction of squared-error loss.

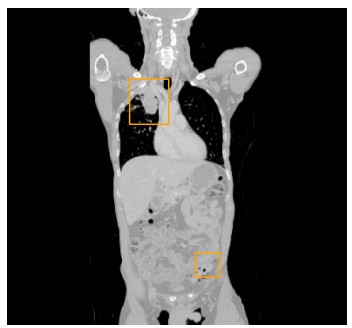 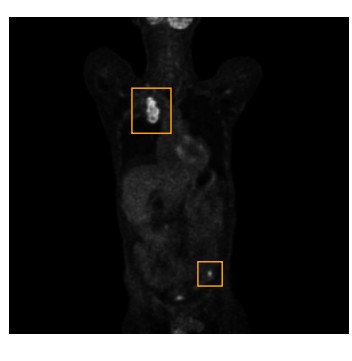 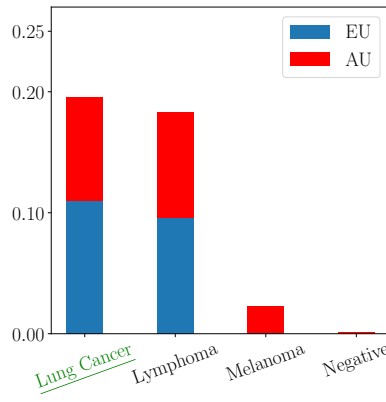

Figure 2: Coronal 2D image from a patient with malignant lesions. Left: CT, middle: PET with *segmentation* by medical experts, right: corresponding *aleatoric* and *epistemic* uncertainties, *ground truth class* and *predicted class*.

Summing over all label-wise uncertainties yields global measures of total, aleatoric, and epistemic uncertainty:

$$\text{TU}(Q) \coloneqq \sum_{k=1}^{K} \text{Var}(Y_k) \qquad (20)$$

$$\text{AU}(Q) \coloneqq \sum_{k=1}^{K} \mathbb{E}[\Theta_k(1 - \Theta_k)] \qquad (21)$$

$$\text{EU}(Q) \coloneqq \sum_{k=1}^{K} \text{Var}(\Theta_k) \qquad (22)$$

Finally we demonstrate that variance-based measures (20), (21), and (22) satisfy a number of desirable properties, discussed in depth in Section 2.2.

**Theorem 3.2.** *Variance-based measures* (20), (21), *and* (22) *satisfy Axioms A0, A1, A2 (only for TU), A3 (strict version), A4 (strict version) and A5–A7.*

Let us highlight that, while entropy-based measures do not fulfill property A5 (as previously pointed out by Wimmer et al. [2023]), this property is now met by the variance-based measures.

On a further note, let us remark that the idea of using variance-based uncertainty measures is not completely new. In particular, a decomposition derived from the law of total variance has been used in regression problems for quite some time [Depeweg et al., 2018]. Moreover, Duan et al. [2024] introduce variance-based uncertainty measures for classification, yet they motivate this from the EDL paradigm and do not discuss any theoretical properties.

## 4  EXPERIMENTS

In this experimental section, our aim is twofold. Firstly, we empirically illustrate the practical efficacy of the *label-wise* uncertainty quantification approach, as motivated in the

preceding sections. This is achieved through experiments conducted on medical data sets, where uncertainty quantification is deemed particularly critical. Our results not only reinforce the theoretical underpinnings discussed earlier but also highlight the importance of reliable uncertainty quantification in high-stakes medical applications.

Secondly, additional experiments on common benchmark data sets are designed to illustrate that adopting a label-wise perspective does not come at the expense of a *global* viewpoint. Due to the fundamental lack of a ground truth in studying uncertainty (as opposed to predictive performance where ground-truth labels are usually available), it is challenging to assess the quality of the uncertainty estimates.

As such, we study the (global) effectiveness of the proposed measures in two different tasks: prediction with abstention and out-of-distribution (OoD) detection.

Details on model architecture and training setup as well as additional experiments can be found in Appendix B and Appendix C, respectively. The code is available in a public repository (`https://github.com/YSale/label-uq`).

### 4.1  LABEL-WISE UNCERTAINTIES

For the evaluation in the medical domain, we use a data set of Positron Emission Tomography/Computed Tomography (PET/CT) images which comprises 501 full-body scans from patients with malignant lymphoma, melanoma, and lung cancer, as well as 513 scans from individuals without malignant lesions (negative controls) [Gatidis et al., 2022]. Each scan is annotated with a tumor segmentation performed by a medical expert. We extract from each 3D CT and PET volume multiple 2D images which are used to train a deep neural network ensemble and evaluate the label-wise uncertainty quantification.

Figure 2 depicts a qualitative example of a 2D PET/CT

image from the data set with the corresponding label-wise uncertainties from our evaluation. We observe low aleatoric uncertainty and negligible epistemic uncertainty for the melanoma class. This implies that the approximation of the aleatoric uncertainty is reliable. On the contrary, the classes lung cancer and lymphoma are associated with high aleatoric and high epistemic uncertainty, suggesting that the prediction with respect to these classes may not be accurate. This observation is plausible from a medical perspective as we observe multiple tumors in the image which are not limited to the lung area and thus might indicate a different tumor class as well. In this instance, we could request a medical expert to annotate additional data for the classes lung cancer and lymphoma, thereby diminishing the epistemic uncertainty associated with these classes. Here, a global measure of uncertainty would only give epistemic uncertainty with respect to all classes, meaning a doctor would have to annotate data for all classes.

Having a more detailed understanding of the label-wise uncertainties is crucial for the decision-making in medical applications supporting a given diagnosis. Moreover, it allocates resources to the relevant classes and saves valuable examination time and costs.

In Appendix C.1, we provide further examples of images with the highest total, aleatoric, and epistemic uncertainty.

## 4.2 ACCURACY-REJECTION CURVES

We generate Accuracy-Rejection Curves (ARCs) by rejecting the predictions for instances on which the predictor is most uncertain and computing the accuracy on the remaining subset [Hühn and Hüllermeier, 2009]. Given a good uncertainty quantification method, the accuracy should monotonically increase with the percentage of rejected instances, because the model misclassifies instances with low uncertainty less often than instances with high uncertainty.

To approximate the second-order distribution, we train an ensemble of five neural networks on the CIFAR10 data set [Krizhevsky et al., 2009]. We compare the proposed label-wise entropy- and variance-based uncertainty measures to the entropy-based baseline (cf. Section 2.1) as used in the Bayesian setting. Figure 3 shows the ARCs for the CIFAR10 data set. The accuracies are reported as the mean over five independent runs and the standard deviation is depicted by the shaded area.

The ARCs for all uncertainty measures closely align with the entropy-based baseline and exhibit similar qualitative behaviors. This highlights the effectiveness of the global measure derived from the local (label-wise) measures.

In this regard, let us note that our goal is *not* to demonstrate that label-wise constructed measures always perform better than their entropy equivalents. Instead, we show that they

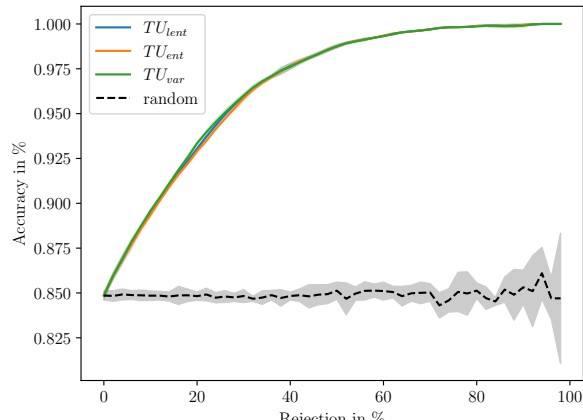

Figure 3: Accuracy-rejection curves generated on the CIFAR10 data set. We compare entropy ($TU_{ent}$), label-wise constructed entropy ($TU_{lent}$), and the variance-based ($TU_{var}$) measure of total uncertainty.

fulfill many desirable properties and are highly competitive in downstream task applications. In particular, this is a very relevant use-case in medical scenarios as it implies that a high classification accuracy can be reached by rejecting a portion of the data. In practice, this could mean that a machine learning algorithm, supplying predictions for unambiguous cases, can work in tandem with a medical expert, examining images deemed too difficult for machine-based prediction, to achieve high accuracy overall.

Further experiments using medical and machine learning data sets can be found in Appendix C.

## 4.3 OUT-OF-DISTRIBUTION DETECTION

Another way to assess and compare measures of uncertainty is to conduct out-of-distribution detection experiments. We train a model on an in-distribution (iD) data set, and compute uncertainty values on instances of the iD test set. Subsequently, the model is exposed to data from an OoD data set, and we similarly assess the uncertainty for these new instances. The model, which has not previously encountered the OoD data, is expected to exhibit increased epistemic uncertainty for these instances. The ability to distinguish between iD and OoD data is an important requirement for a reliable machine learning model, because accurate predictions cannot be guaranteed on OoD data.

Our approach involves training an ensemble of five neural networks on the FashionMNIST [Xiao et al., 2017] data set (iD), using MNIST [LeCun et al., 1998] and KMNIST [Clanuwat et al., 2018] as our chosen OoD data sets. To determine the effectiveness in distinguishing between iD and OoD instances using total uncertainty, we calculate the AUROC and compute its mean and standard deviation across five independent runs. Similarly, we conduct OoD experiments for CIFAR10 (iD) with SVHN [Netzer et al.,

Table 1: OoD detection performance. `AUROC` and standard deviation over five runs are reported. $EU_{ent}$ denotes mutual information, $EU_{var}$ the variance-based measure, and $EU_{lent}$ label-wise entropy. Best performance is in **bold**.

| | FashionMNIST | | CIFAR10 | |
|---|---|---|---|---|
| Methods | MNIST | KMNIST | SVHN | CIFAR10.2 |
| $EU_{var}$ | $.882 \pm 0.18$ | $.959 \pm .005$ | $\mathbf{.761 \pm .022}$ | $.999 \pm .001$ |
| $EU_{lent}$ | $.894 \pm .017$ | $.967 \pm .004$ | $.745 \pm .027$ | $\mathbf{1.00 \pm .001}$ |
| $EU_{ent}$ | $\mathbf{.895 \pm .017}$ | $\mathbf{.969 \pm .004}$ | $.760 \pm .026$ | $.998 \pm .002$ |

2011] and CIFAR10.2 [Lu et al., 2020] as OoD data sets.

Table 1 shows the results for the networks trained on FashionMNIST or CIFAR10. Overall, the compared measures perform well. In particular, the label-wise measures and the entropy-based measures yield similar results, emphasizing again that using the label-wise measures in a global way does not sacrifice performance.

## 5 CONCLUDING REMARKS

We introduced a novel approach to quantifying uncertainty in classification tasks, adopting a label-wise perspective that allows for reasoning about uncertainty at the level of individual classes. This can be beneficial for decision-making in situations where incorrect predictions have unequal consequences for different classes, and for deciding about the right course of action to reduce uncertainty. Addressing criticisms in the recent literature and problems of the commonly used information-theoretic (entropy-based) measures, we showed that our measures satisfy many desirable properties. We also presented empirical results highlighting the practical usefulness of these measures. Overall, we trust that this work is a step towards a more interpretable representation of uncertainty that will be beneficial for safety-critical applications.

Our approach to decomposing uncertainty in a label-wise manner is in direct correspondence with the one-vs-rest decomposition of a multinomial into several binary classification problems. An interesting idea for future work, therefore, is the use of other decomposition techniques. In any case, thanks to the binarization, our approach is amenable to a very broad class of uncertainty measures. Specifically, we presented a framework that is parameterized by a loss function (proper scoring rule) $\phi$. As another direction of future work, we plan to elaborate more deeply on the appropriate choice of this loss, and the effect it has on uncertainty quantification.

## Acknowledgements

Yusuf Sale and Lisa Wimmer are supported by the DAAD program Konrad Zuse Schools of Excellence in Artificial Intelligence, sponsored by the Federal Ministry of Education and Research.

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

# Label-wise Aleatoric and Epistemic Uncertainty Quantification
## (Supplementary Material)

**Yusuf Sale**[1,3]    **Paul Hofman**[1,3]    **Timo Löhr**[1,3]    **Lisa Wimmer**[2,3]    **Thomas Nagler**[2,3]    **Eyke Hüllermeier**[1,3]

[1]Institute of Informatics, LMU Munich, Munich, Germany
[2]Department of Statistics, LMU Munich, Munich, Germany
[3]Munich Center for Machine Learning

## A   PROOFS

**Proposition A.1.** *Let $Q' \in \Delta_K^{(2)}$ be a mean-preserving spread of $Q \in \Delta_K^{(2)}$, i.e., let $\Theta \sim Q$, and $\Theta' \sim Q'$ be two random vectors such that we have $\Theta' \stackrel{d}{=} \Theta + Z$, for some random vector $Z$ with $\mathbb{E}[Z \,|\, \sigma(\Theta)] = 0$ almost surely.*

*Now, define*

$$EU(Q) = \mathbb{E}_Q[\mathrm{D_{KL}}(\Theta \,\|\, \bar{\theta})], \tag{23}$$

*where $\mathrm{D_{KL}}(\Theta \,\|\, \bar{\theta})$ denotes the Kullback-Leibler (KL) divergence of $\Theta$ from its mean $\bar{\theta}$. Then the claim is that $EU(Q') > EU(Q)$.*

*Proof.* First, note that $\mathrm{D_{KL}}(\Theta \,\|\, \bar{\theta})$ is a convex function of $\Theta$ since $\bar{\theta} \in \Delta_K$ is a constant. Given that $\Theta' \stackrel{d}{=} \Theta + Z$ and $\mathbb{E}[Z \,|\, \sigma(\Theta)] = 0$ almost surely, it follows that $\mathbb{E}_{Q'}[\Theta' \,|\, \sigma(\Theta)] = \Theta$.

Jensen's inequality states that for a strict convex function $f$ and a non-constant random vector $Y$, we have $\mathbb{E}[f(Y)] > f(\mathbb{E}[Y])$. Then Jensen's inequality implies:

$$\mathbb{E}_{Q'}[\mathrm{D_{KL}}(\Theta' \| \bar{\theta}) \,|\, \sigma(\Theta)] > \underbrace{\mathrm{D_{KL}}(\mathbb{E}_{Q'}[\Theta' \,|\, \sigma(\Theta)] \,\|\, \bar{\theta})}_{=\mathrm{D_{KL}}(\Theta \,\|\, \bar{\theta})} \quad \text{a.s.}$$

By this we get

$$\mathbb{E}_{Q'}[\mathrm{D_{KL}}(\Theta' \,\|\, \bar{\theta})] = \mathbb{E}_Q[\mathbb{E}_{Q'}[\mathrm{D_{KL}}(\Theta' \,\|\, \bar{\theta}) \,|\, \sigma(\Theta)]] > \mathbb{E}_Q[\mathrm{D_{KL}}(\Theta \,\|\, \bar{\theta})], \tag{24}$$

by the law of total expectation. This concludes the proof. □

*Proof of Theorem 3.1.* We prove that the entropy-based uncertainty measures (17), (18), and (19) satisfy Axioms A0, A1, A2 (only for TU), A3 (strict version), A4 (strict version, only for TU), A6, and A7 of Section 2.2.

A0: This property holds true since entropy $H(\cdot)$ and KL-divergence $D_{KL}(\cdot \,||\, \cdot)$ are non-negative.

A1: Let $Q = \delta_{\boldsymbol{\theta}} \in \Delta_K^{(2)}$ be a Dirac measure on $\boldsymbol{\theta} \in \Delta_K$. Since $D_{KL}(\theta_k \,||\, \bar{\theta}_k)] = 0$ if and only if $\theta_k = \bar{\theta}_k$ and consequently $\mathbb{E}_{Q_k}[D_{KL}(\Theta_k \,||\, \bar{\theta}_k)] = 0$ for all $k \in \{1, \ldots, K\}$ the claim holds true.

A2: First we show that the function $U(\boldsymbol{\theta}) = \sum_{k=1}^{K} H(\theta_k)$ is strictly concave on $\Delta_K$ with unique maximizer $\boldsymbol{\beta} = (1/K, \ldots, 1/K)$. We observe that

$$
\nabla^2 U(\boldsymbol{\theta}) = \begin{bmatrix}
- \left[ \frac{\log(2)}{\theta_1} + \frac{\log(2)}{1-\theta_1} \right] & 0 & \cdots & 0 \\
0 & - \left[ \frac{\log(2)}{\theta_2} + \frac{\log(2)}{1-\theta_2} \right] & \cdots & 0 \\
\vdots & \vdots & \ddots & \vdots \\
0 & 0 & \cdots & - \left[ \frac{\log(2)}{\theta_K} + \frac{\log(2)}{1-\theta_K} \right]
\end{bmatrix}
$$

is negative definite. To find the maximizer, we consider the Lagrangian dual:

$$
\max_{\boldsymbol{\theta} \in \Delta_k, \lambda} U^*(\boldsymbol{\theta}, \lambda) = \max_{\boldsymbol{\theta} \in \Delta_k, \lambda} U(\boldsymbol{\theta}) + \lambda \left( \sum_{k=1}^{K} \theta_k - 1 \right).
$$

The first-order conditions are

$$
\frac{\partial U^*(\boldsymbol{\theta}, \lambda)}{\partial \theta_k} = \log_2(\theta_k) - \log_2(1 - \theta_k) + \lambda = 0, \ k \in \{1, ..., K\} \text{ and } \sum_{k=1}^{K} \theta_k = 1,
$$

which are solved by $\boldsymbol{\theta} = (1/K, \ldots, 1/K)$ and $\lambda = \log_2 \frac{K-1}{K} - \log_2 \frac{1}{K}$. This implies that TU is maximized for $Q \in \Delta_K^{(2)}$ such that $\mathbb{E}[\Theta_k] = 1/K$ for all $k \in \{1, \ldots, K\}$. The latter holds true for $Q$ being the uniform distribution on $\Delta_K^{(2)}$.

Thus, for any $Q \in \Delta_K^{(2)}$ satisfying $\mathbb{E}[\Theta_k] = 1/K$ for all $k \in \{1, \ldots, K\}$ we obtain

$$
TU(Q) = \log_2(K) + (K-1) \log_2(K/(K-1)).
$$

It is easy to show that the maximum of EU aligns with that of TU. Assume, for the sake of argument, that EU is maximal for $Q_{\text{Unif}}$ being the uniform distribution on $\Delta_K^{(2)}$. Since TU decomposes additively in AU and EU, it follows that $AU(Q_{\text{Unif}}) = 0$. However, given that $AU(Q_{\text{unif}}) > 0$, this leads to a contradiction, demonstrating that EU cannot be maximal for $Q_{\text{Unif}}$.

A3: Let $Q' \in \Delta_K^{(2)}$ be a mean-preserving spread of $Q \in \Delta_K^{(2)}$, i.e., let $\Theta \sim Q, \Theta' \sim Q'$ be two random vectors such that $\Theta' \overset{d}{=} \Theta + Z$, for some random vector $Z$ with $\mathbb{E}[Z \,|\, \sigma(\Theta)] = 0$ almost surely. Applying Proposition A.1 yields $\mathbb{E}_{Q'_k}[D_{KL}(\Theta'_k \,||\, \bar{\theta}_k)] > \mathbb{E}_{Q_k}[D_{KL}(\Theta_k \,||\, \bar{\theta}_k)]$ for all $k \in \{1, \ldots, K\}$ and with that $EU(Q') > EU(Q)$.

Since we have by definition $TU(Q') = \sum_{k=1}^{K} H(\mathbb{E}_{Q'_k}[\Theta'_k])$, and $\mathbb{E}_{Q'_k}[\Theta'_k] = \mathbb{E}_{Q_k}[\Theta_k]$ for all $k \in \{1, \ldots, K\}$ by the mean-preserving spread assumption, $TU(Q') = TU(Q)$ follows.

A4: Let $Q'$ be a spread-preserving center shift of $Q$ such that $\mathbb{E}[\Theta'] = \lambda \mathbb{E}[\Theta] + (1 - \lambda)(1/K, \ldots, 1/K)^\top$ for some $\lambda \in (0, 1)$. Because also $\Theta' = \Theta + z$ with $z \neq 0$, this implies $\mathbb{E}[\Theta] \neq (1/K, \ldots, 1/K)^\top$. From the proof of A2 we know that $(1/K, \ldots, 1/K)$ maximizes $\sum_{k=1}^{K} H(\theta_k)$ and concavity of $H(\cdot)$ implies $TU(Q') > TU(Q)$.

A6: Let $\delta_m \in \Delta_{\delta_m}$, such that we have

$$
AU(\delta_m) = \sum_{k=1}^{K} \mathbb{E}[H(Y_k \,|\, \Theta_k)]
$$

$$
= \sum_{k=1}^{K} \sum_{y_k \in \{0,1\}} H(\delta_{y_k}) \lambda_{y_k}(\delta_{y_k})
$$

$$
= 0,
$$

where the last equation holds true, since $H(\delta_{y_k}) = 0$ for all $y_k \in \{0, 1\}$ and $k \in \{1, \ldots, K\}$.

A7: Let $Q \in \Delta_K^{(2)}$ and denote by $Q_{|\mathcal{Y}_1}$ and $Q_{|\mathcal{Y}_2}$ the corresponding marginalized distribution, where $\mathcal{Y}_1$ and $\mathcal{Y}_2$ are partitions of $\mathcal{Y}$. It holds

$$\mathrm{TU}_{\mathcal{Y}}(Q) = \sum_{k \in \mathcal{Y}} \mathrm{H}(\mathbb{E}_{Q_k}[\Theta_k]) = \sum_{k \in \mathcal{Y}_1} \mathrm{H}(\mathbb{E}_{Q_k}[\Theta_k]) + \sum_{k \in \mathcal{Y}_2} \mathrm{H}(\mathbb{E}_{Q_k}[\Theta_k])$$

$$= \mathrm{TU}_{\mathcal{Y}_1}(Q_{|\mathcal{Y}_1}) + \mathrm{TU}_{\mathcal{Y}_2}(Q_{|\mathcal{Y}_2}),$$

similarly the same holds for AU. Due to the additive decomposition the claim is also true for EU.

This concludes the proof. $\qquad\square$

*Proof of Theorem 3.2.* We prove that variance-based uncertainty measures (20), (21), and (22) satisfy Axioms A0, A1, A2 (only for TU), A3 (strict version), A4 (strict version) and A5–A7 of Section 2.2.

A0: This property holds trivially true.

A1: Let $Q = \delta_{\boldsymbol{\theta}} \in \Delta_K^{(2)}$ be a Dirac measure on $\boldsymbol{\theta} \in \Delta_K$. Then $\mathrm{EU}[\delta_{\boldsymbol{\theta}}] = 0$ holds trivially true, since $\mathrm{Var}_{Q_k}[\Theta_k] = 0$ for all $k \in \{1, \ldots, K\}$. The other direction follows similarly.

A2: First we show that the function $V(\boldsymbol{\theta}) = \sum_{k=1}^{K} \theta_k(1 - \theta_k)$ is strictly concave on $\Delta_K$ with unique maximizer $\boldsymbol{\beta} = (1/K, \ldots, 1/K)$. It holds

$$\nabla^2 V(\boldsymbol{\theta}) = -2\,\mathrm{diag}(1, \ldots, 1),$$

which is negative definite. To find the maximizer, we consider the Lagrangian dual:

$$\max_{\boldsymbol{\theta} \in \Delta_K, \lambda} V^*(\boldsymbol{\theta}, \lambda) = \max_{\boldsymbol{\theta} \in \Delta_K, \lambda} V(\boldsymbol{\theta}) + \lambda\left(\sum_{k=1}^{K} \theta_k - 1\right).$$

The first-order conditions are

$$\frac{\partial V^*(\boldsymbol{\theta}, \lambda)}{\partial \theta_k} = 1 - 2\theta_k + \lambda = 0, \ k \in \{1, \ldots, K\} \text{ and } \sum_{k=1}^{K} \theta_k = 1,$$

which are solved by $\boldsymbol{\theta} = \boldsymbol{\beta}$ and $\lambda = -(K-2)/K$. This implies that TU is maximized for any $Q \in \Delta_K^{(2)}$ such that $\mathbb{E}[\Theta_k] = 1/K$ for all $k \in \{1, \ldots, K\}$. The latter holds true for $Q$ being the uniform distribution on $\Delta_K^{(2)}$.

The proof that EU is not maximal for $Q_{\mathrm{Unif}}$ being the uniform distribution on $\Delta_K^{(2)}$ is completely analogous to the proof of A3 in Theorem 3.1.

A3: Let $Q' \in \Delta_K^{(2)}$ be a mean-preserving spread of $Q \in \Delta_K^{(2)}$, i.e., let $\boldsymbol{\Theta} \sim Q, \boldsymbol{\Theta}' \sim Q'$ be two random vectors such that $\boldsymbol{\Theta}' \overset{d}{=} \boldsymbol{\Theta} + \boldsymbol{Z}$, for some random vector $\boldsymbol{Z}$ with $\mathbb{E}[\boldsymbol{Z} \,|\, \sigma(\boldsymbol{\Theta})] = 0$ almost surely. Then, we have the following:

$$\mathrm{EU}(Q') = \sum_{k=1}^{K} \mathrm{Var}(\Theta_k + Z_k) \tag{25}$$

$$= \sum_{k=1}^{K} \mathrm{Var}(\Theta_k) + \mathrm{Var}(Z_k) + 2\,\mathrm{Cov}(\Theta_k, Z_k) \tag{26}$$

$$= \mathrm{EU}(Q) + \sum_{k=1}^{K} \mathrm{Var}(Z_k) \tag{27}$$

$$> \mathrm{EU}(Q) \tag{28}$$

Note that the equality (27) holds true, since we know that $\mathrm{Cov}(\Theta_k, Z_k) = 0$. To see this, observe that we have $\mathbb{E}[\Theta_k Z_k] = \mathbb{E}[\mathbb{E}[\Theta_k Z_k \,|\, \sigma(\Theta_k)]] = 0$ due to the mean-preserving spread assumption. Similarly, we know that $\mathbb{E}[Z_k] = 0$, such that we have $\mathrm{Cov}(\Theta_k, Z_k) = \mathbb{E}[\Theta_k Z_k] - \mathbb{E}[\Theta_k]\mathbb{E}[Z_k] = 0$. The inequality (28) is strict since by assumption $\max_k \mathrm{Var}(Z_k) > 0$.

Since we have $\mathbb{E}[\boldsymbol{\Theta}] = \mathbb{E}[\boldsymbol{\Theta}']$ by assumption, the weak version of A3 holds for TU.

A4: Let $Q'$ be a spread-preserving location shift of $Q$ such that $\mathbb{E}[\boldsymbol{\Theta}'] = \lambda\mathbb{E}[\boldsymbol{\Theta}] + (1 - \lambda)(1/K, \ldots, 1/K)^\top$ for some $\lambda \in (0, 1)$. From the proof of A2 we know that $\boldsymbol{\beta} = (1/K, \ldots, 1/K)$ maximizes $\sum_{k=1}^{K} \theta_k(1 - \theta_k)$, which in turn immediately implies $\mathrm{TU}(Q') > \mathrm{TU}(Q)$. The inequality for AU is then implied by $\mathrm{EU}(Q') = \mathrm{EU}(Q)$ (Axiom A5).

A5: Let $\boldsymbol{\Theta} \sim Q$, and $(\boldsymbol{\Theta} + \boldsymbol{z}) \sim Q'$, where $\boldsymbol{z} \neq \boldsymbol{0}$ is a constant. Then, we observe

$$
\begin{aligned}
\mathrm{EU}(Q') &= \sum_{k=1}^{K} \mathrm{Var}(\Theta_k + z_k) \\
&= \sum_{k=1}^{K} \mathbb{E}[((\Theta_k + z_k) - \mathbb{E}[\Theta_k + z_k])^2] \\
&= \sum_{k=1}^{K} \mathbb{E}[(\Theta_k - \mathbb{E}[\Theta_k])^2] \\
&= \mathrm{EU}(Q).
\end{aligned}
$$

A6: With $\delta_m \in \Delta_{\delta_m}$ we have

$$
\begin{aligned}
\mathrm{AU}(\delta_m) &= \sum_{k=1}^{K} \mathbb{E}[\Theta_k(1 - \Theta_k)] \\
&= \sum_{k=1}^{K} \lambda_k(1 - 1) + (1 - \lambda_k)(0 - 0) \\
&= 0.
\end{aligned}
$$

A7: Let $Q \in \Delta_K^{(2)}$ and further denote by $Q_{|\mathcal{Y}_1}$ and $Q_{|\mathcal{Y}_2}$ the corresponding marginalized distribution, where $\mathcal{Y}_1$ and $\mathcal{Y}_2$ are partitions of $\mathcal{Y}$. It holds

$$
\begin{aligned}
\mathrm{TU}_{\mathcal{Y}}(Q) = \sum_{k \in \mathcal{Y}} \mathrm{Var}(Y_k) &= \sum_{k \in \mathcal{Y}_1} \mathrm{Var}(Y_k) + \sum_{k \in \mathcal{Y}_2} \mathrm{Var}(Y_k) \\
&= \mathrm{TU}_{\mathcal{Y}_1}(Q_{|\mathcal{Y}_1}) + \mathrm{TU}_{\mathcal{Y}_2}(Q_{|\mathcal{Y}_2}),
\end{aligned}
$$

similarly the same holds for AU. Due to the additive decomposition the claim is also true for EU.

This concludes the proof. $\qquad\qquad\square$

# B EXPERIMENTAL DETAILS

In this section, we provide a detailed overview of the experimental setup to allow reproduction of the results.

The experimental code is written in `Python 3.9` using the `PyTorch` [Paszke et al., 2019] library.

## B.1 EXPERIMENTS ON ML DATA SETS

**Data sets.** For all data sets, we use the respective dedicated train-test splits. We only use pre-processing for the CIFAR10 data set. Each image is normalized using the mean and standard deviation per channel of the training set. Additionally, the training images are cropped randomly (while adding 4 pixels of padding on every border and randomly flipped horizontally).

**Ensembles.** The ensembles are built using two base models: a Convolutional Neural Network (`CNN`) and a `ResNet18` [He et al., 2016]. The `CNN` has two convolutional layers followed by two fully connect layers. The convolutional layers have 32 and 64 filters of 5 by 5 and the fully connected layers have 512 and 10 neurons, respectively. The layers have `ReLU` activations and the last layer uses a softmax function to output probabilities. The `ResNet18` model has a fully connected last layer of 10 units and a softmax function to generate probabilities for 10 classes. The output of the ensemble is generated by averaging over the outputs of the individual ensemble members.

### Accuracy-Rejection Curves

We train 5 `CNNs` on FMNIST, MNIST and KMNIST and 5 `ResNets` on CIFAR10 and SVHN. We use `Adam` [Kingma and Ba, 2015] with the default hyper-parameters to train the `CNNs` in 10 epochs for MNIST and 20 epochs for FMNIST and KMNIST using a batch size of 256. We train the `ResNets` using stochastic gradient descent with weight decay set to $10^{-4}$, momentum at 0.9, and a learning rate of 0.1, setting the learning rate to 0.001 at epoch 20 and to 0.0001 at epoch 25. The models are trained for 30 epochs in total. The ARCs are then generated using the test set.

### Out-of-Distribution Detection

We train 5 `CNNs` on FashionMNIST and 5 `ResNets` on CIFAR10, using the same setup as for the ARCs. Epistemic uncertainty is computed on the test sets of the corresponding data sets without applying any data augmentation.

## B.2 EXPERIMENTS ON MEDICAL IMAGES

**Data set.** We use a publicly available PET/CT image data set [Gatidis et al., 2022] with respective dedicated train-test splits. During the preprocessing, we extract 2D images as coronal slices from the 3D PET and CT image volumes. Furthermore, each image is resized to $400 \times 400$ pixels, normalized using the mean and standard deviation per channel of the training set and stacked into a three channel image consisting of a PET, CT, and fusion channel. The final data set consists of 96000 2D images coming from 1014 patients.

**Ensembles.** The ensembles are built using a `ResNet50` [He et al., 2016]. The `ResNet50` model has a fully connected last layer of 4 units and a softmax function to generate probabilities for 4 classes. The output of the ensemble is generated by averaging over the outputs of the individual ensemble members. We use Adam [Kingma and Ba, 2015] to train each model with a cross entropy loss for 40 epochs with a learning rate of 0.001 and a batch size of 50. All models are evaluated on a separate test set.

### Accuracy-Rejection Curves

We train 5 `ResNets` on the PET/CT image data set. We use Adam [Kingma and Ba, 2015] to train each model with a cross entropy loss for 40 epochs with a learning rate of 0.001 and a batch size of 50. The ARCs are then generated using the test set.

# C ADDITIONAL RESULTS

In this section, we report on experiments that we perform in addition to the ones presented in the main paper.

## C.1 LABEL-WISE UNCERTAINTIES

Figure 4 presents additional medical images with the highest total, aleatoric, and epistemic uncertainties. Similarly, Figure 5 showcases images with the highest total, aleatoric, and epistemic uncertainties for the MNIST data set.

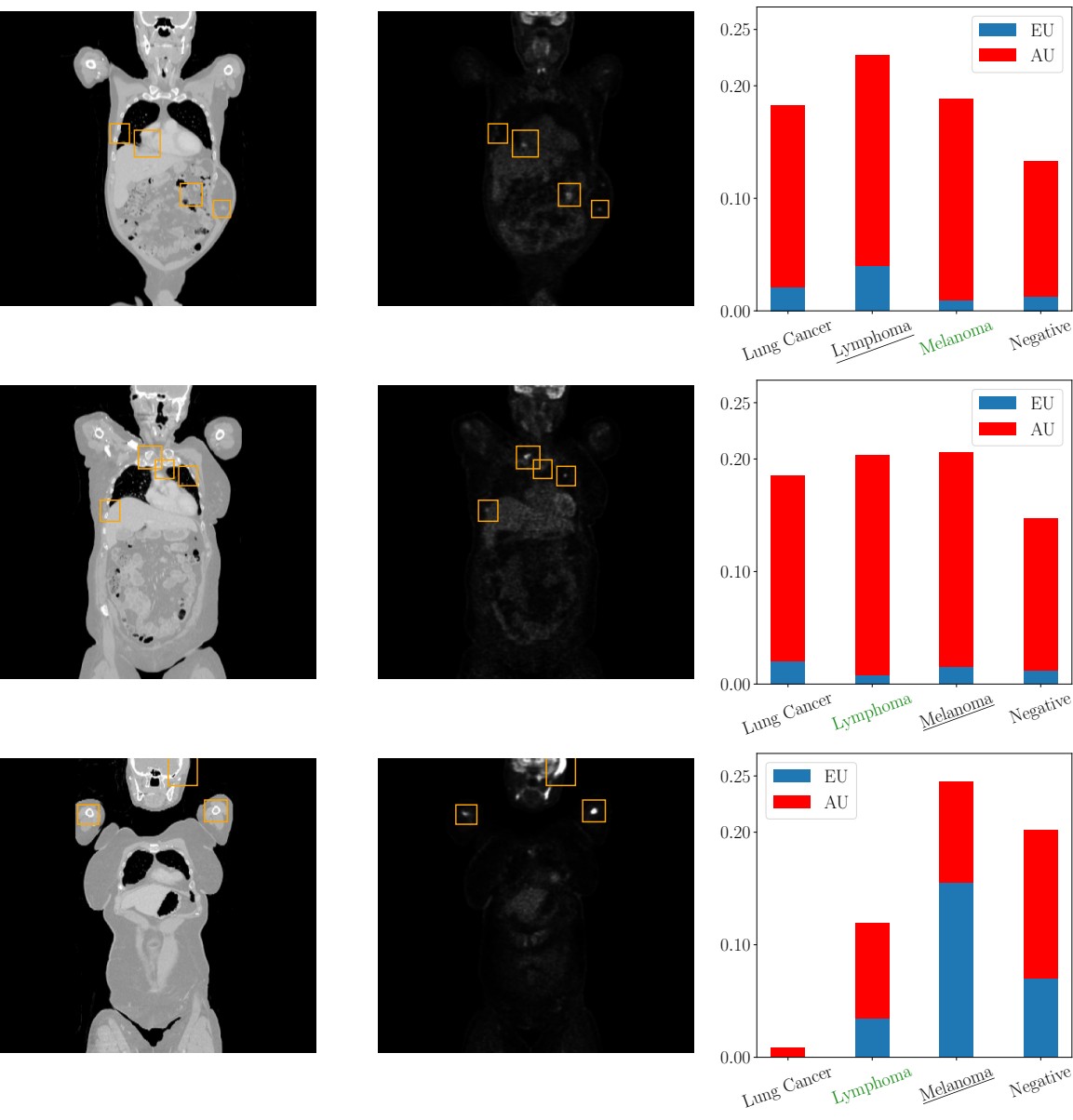

Figure 4: Medical image examples with highest total *(top)*, *aleatoric (middle)* and *epistemic* uncertainties *(bottom)* along with their corresponding label-wise uncertainties, *ground truth class* and *predicted class*.

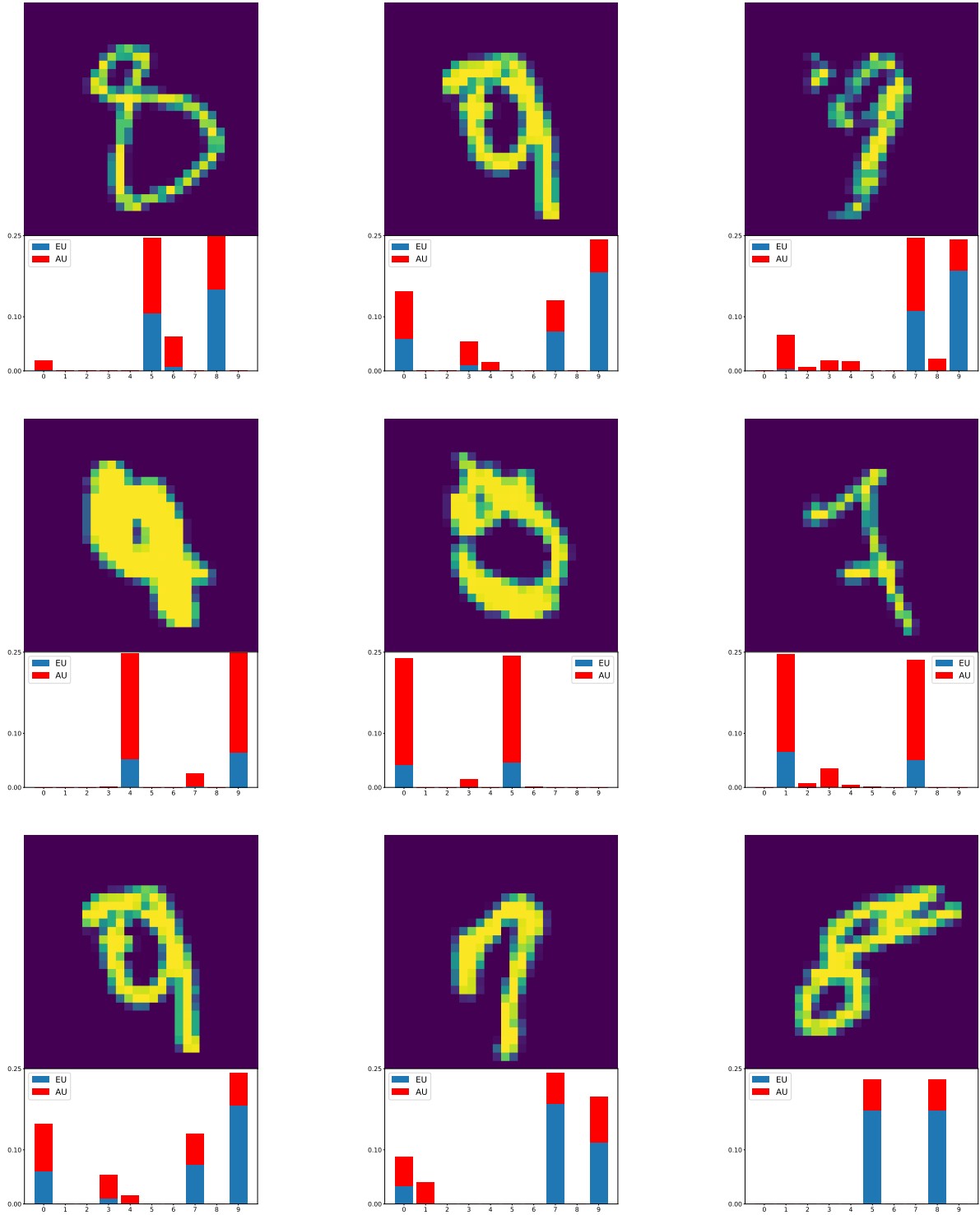

Figure 5: MNIST instances with highest total *(top)*, *aleatoric (middle)*, and *epistemic* uncertainties *(bottom)* along with their corresponding label-wise uncertainties.

## C.2 ACCURACY-REJECTION CURVES

We train an ensemble of 5 neural networks on the data sets using the setup outlined in Section B. Figure 6 shows the accuracy-rejection curves for the medical data and the FMNIST data set. The accuracies are reported as the mean over five runs and the standard deviation is depicted by the shaded area.

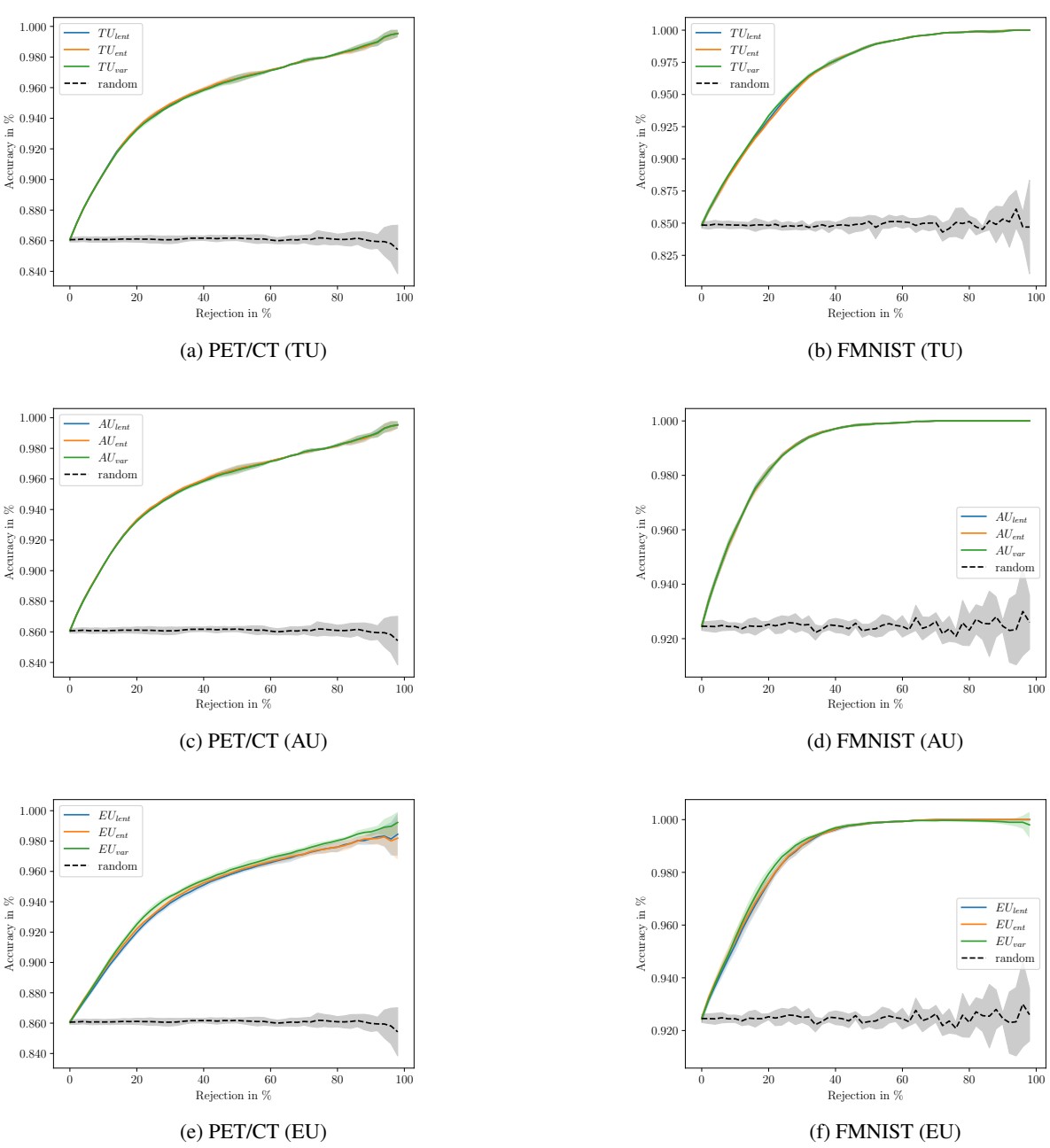

Figure 6: Accuracy-rejection curves on PET/CT *(left)* and FMNIST *(right)*.

We observe results consistent with those presented in the main paper. The label-wise measures exhibit behavior very similar to the usual entropy measures, with most measures increasing monotonically.

## C.3   HOLDOUT EXPERIMENTS

**Experimental Details**

**Data sets.** We perform experiments on CIFAR10 and FMNIST data sets with corresponding train-test splits. We only use pre-processing for the CIFAR10 data set. Each image is normalized using the mean and standard deviation per channel of the training set. Additionally, the training images are cropped randomly (while adding 4 pixels of padding on every border and randomly flipped horizontally).

**Ensemble.** The ensembles are built using two base models: a Convolutional Neural Network (`CNN`) and a `ResNet18` [He et al., 2016]. The `CNN` has two convolutional layers followed by two fully connect layers. The convolutional layers have 32 and 64 filters of 5 by 5 and the fully connected layers have 512 and 10 neurons, respectively. The layers have `ReLU` activations and the last layer uses a softmax function to output probabilities. The `ResNet18` model has a fully connected last layer of 10 units and a softmax function to generate probabilities for 10 classes. The output of the ensemble is generated by averaging over the outputs of the individual ensemble members.

**Experiments.** For the CIFAR10 data set, we train the ensemble for 20 epochs on a small subset of the train data (10% of the initial train data, the other 90% is reserved as holdout). To prevent class imbalance, we remove instances with the highest EU class (EU per class is computed by first calculating the label-wise EU for all instances using the variance-based EU measure, and then averaging over instances from the same class) from the train data, and add the same amount of holdout data (from the class, which was identified after the initial 20 epochs training as highest EU class). Although the amount of data for each class remains the same across epochs, the learner is progressively exposed to a broader range of examples from the class with highest EU. In other words, the approach effectively increases the total number of observations the learner encounters from that class over time without leading to class imbalance. This step is repeated for 20 epochs of *continued* learning to ensure the model is trained on a diverse set of examples. Finally, we compare the epistemic uncertainty (of both the class with highest EU and the average of all other classes) *before* and *after* giving the learner access to more data from the class with highest EU. We follow the same procedure for the FMNIST dataset (see executed configurations).

Executed configurations:

(i)  For the CIFAR10 data set:
- The experiment was run with 20 epochs of *initial* training and 20 epochs of *continued* training.
- A hold-out rate of 90% was applied, indicating that a large portion of the data was initially withheld.
- The experiment was repeated for 5 runs.

(ii)  For the FMNIST data set:
- The experiment was run with 5 epochs of *initial* training and 5 epochs of *continued* training.
- A hold-out rate of 99.5% was used, indicating that a large portion of the data was initially withheld.
- The experiment was repeated for 5 runs.

**Experimental Results**

In Table 2 we present both the absolute and relative changes in the EU values for each dataset. Additionally, we include the changes in EU for other classes, with the average being reported. For comparison purposes, we also provide the absolute and relative changes in EU for the class experiencing the second-largest reduction in the EU values ("Next highest drop").

|  | FMNIST | | | CIFAR10 | | |
|---|---|---|---|---|---|---|
|  | Max. EU class | Other classes | Next highest drop | Max. EU class | Other classes | Next highest drop |
| Absolute | **0.0070 ± 0.0006** | 0.0018 ± 0.0000 | 0.0029 ± 0.0000 | **0.0057 ± 0.0011** | 0.0023 ± 0.0000 | 0.0031 ± 0.0000 |
| Relative | **0.7934 ± 0.0616** | 0.3872 ± 0.0000 | 0.5619 ± 0.0000 | **0.5815 ± 0.0304** | 0.3682 ± 0.0000 | 0.4630 ± 0.0000 |

Table 2: Absolute and relative changes in EU.

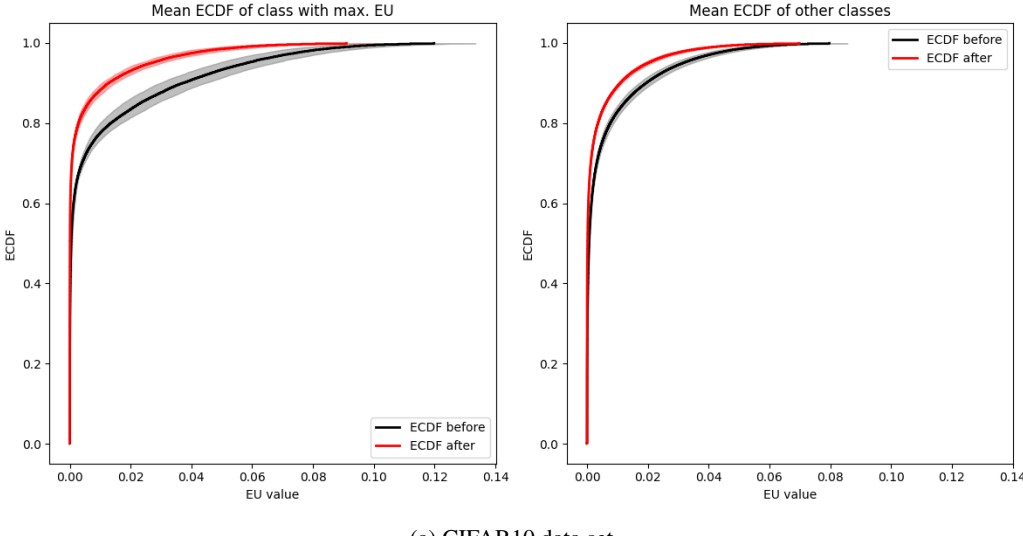

(a) CIFAR10 data set.

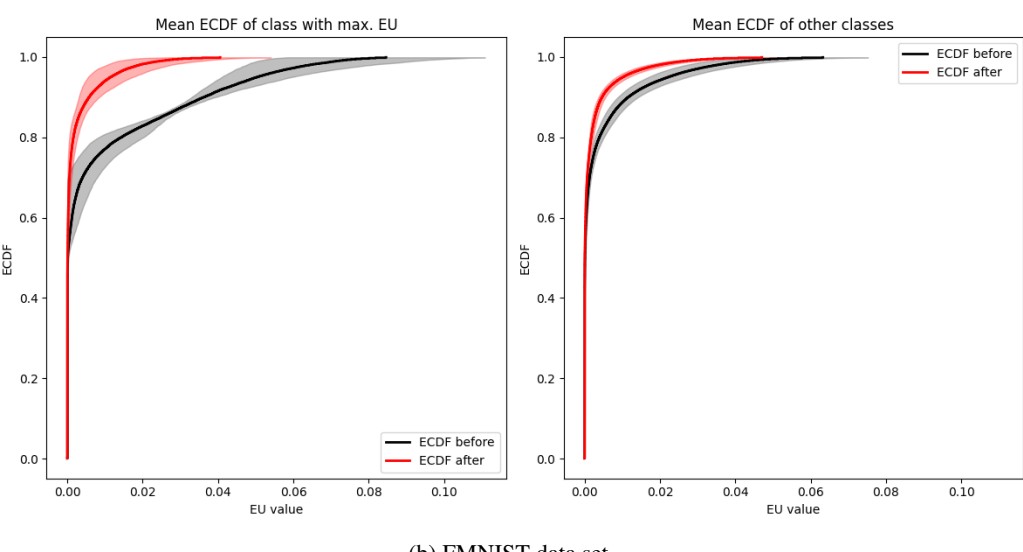

(b) FMNIST data set.

Figure 7: ECDF of class with maximal EU *(left)* and ECDF of other classes *(right)*.

Figure 7 shows the empirical cumulative distribution function (ECDF), averaged over 5 runs of the experiment for the EU values that we observe. On the left we see the ECDF for the class that we identified as having the "highest EU" after the *initial* training, and on the right the averaged ECDF of the "other classes".

**Conclusion.** We conclude that providing the learner with more data from the highest EU class decreases EU for this class the most. While EU for other classes will not remain necessarily constant, it is also important to note that EU is also *not* increasing for other classes.