# OpenReview forum: "Label-wise Aleatoric and Epistemic Uncertainty Quantification"
_auai.org/UAI/2024/Conference — UAI 2024 poster_

### Official Review · Reviewer_fnH4 · 2024-03-15

**Q2-1 Originality-Novelty:** 3
**Q2-2 Correctness-Technical Quality:** 3
**Q2-5 Clarity Of Writing:** 4

**Q1 Summary And Contributions:**

This paper proposes a new definition of the uncertainty quantification for classification task defined label-wise instead of global definitions usually performed. By doing so, the authors aim at giving a more precise understanding of the sources of uncertainty which is critical for some applications such as medicine. For instance if the Epistemic Uncertainty is high, it means that the model requires more data to take a better decision, but if we are able to point for which class it is high, instead of assessing that it is globally high, we can make this data acquisition more efficient in time and in cost, as the costs associated to some classes might be higher than others.

The authors recall and propose two new properties the decomposition in Total Uncertainty, Aleatoric Uncertainty and Epistemic Uncertainty should satisfy. They then establish their contribution by defining label-wise uncertainty quantification using a loss function, which allow to capture entropy and variance based measures. These two measures satisfy the required properties, except one for entropy based, problem that was point out in the literature.

Finally, these label-wise measures are tested experimentally and show similar behaviors than usual ones on the global uncertainty measurement while showing useful on the new focus on label-wise establishment.

**Q2-3 Extent To Which Claims Are Supported By Evidence:**

4: Excellent: all claims are supported by very convincing evidence (in the form of comprehensive experimental evaluation, rigorous mathematical proofs, detailed (pseudo-)code, precise references, well-motivated and realistic assumptions) and the authors deliver what they promise.

**Q2-4 Reproducibility:**

4: Excellent: key resources (e.g. proofs, code, data) are available and key details (e.g. proof sketches, experimental setup) are comprehensively described for competent researchers to confidently and easily reproduce the main results.

**Q3 Main Strengths:**

The paper is well written and the proofs are well founded.
The concept of variance based measurement is interesting and address some concerns of the usual entropy-based measures.

**Q4 Main Weakness:**

I am not an expert of these tools but the proofs seems to be without flaws.
For the experiments I wonder how uncertain are the decisions based on an ensemble of 5 neural networks learnt independently but on the same training data (with different weight initialization and data order within the training data I suppose)

**Q5 Detailed Comments To The Authors:**

At the end of Section 2.1 after Equation (6) you define the Kullback-Leibler divergence which is not present in Equation (6) (only in Section 3.1 of the paper) but I suppose it comes from the definition of EU(Q) given in Equation (23) in Appendix A.

**Q9 Complying With Reviewing Instructions:**

Yes

---

> ### Author Rebuttal · Authors · 2024-04-04
>
> We feel very encouraged by the reviewers' positive comments on our paper.
>
> > For the experiments I wonder how uncertain are the decisions based on an ensemble of 5 neural networks learnt independently but on the same training data (with different weight initialization and data order within the training data I suppose)
>
> The uncertainty depends very much on the concrete ensemble and data at hand. However, we are not exactly sure to understand the question correctly.
> Do you criticize this setting? Maybe you could elaborate a bit more on this point, that would be highly appreciated.
>
> >At the end of Section 2.1 after Equation (6) you define the Kullback-Leibler divergence which is not present in Equation (6) (only in Section 3.1 of the paper) but I suppose it comes from the definition of EU(Q) given in Equation (23) in Appendix A.
>
> While we need KL divergences later on (for entropy-based uncertainty measures), you rightfully point out that this definition is misplaced in the text. Thank you for pointing out, we will streamline accordingly.

---

### Official Review · Reviewer_J46z · 2024-03-16

**Q2-1 Originality-Novelty:** 2
**Q2-2 Correctness-Technical Quality:** 3
**Q2-5 Clarity Of Writing:** 3

**Q1 Summary And Contributions:**

This paper addresses uncertainty quantification in a label-wise perspective for classification problems. Both entropy-based measures and variance-based measures are described and shown to satisfy (some of) desirable properties (A0-A7). Both Aleatoric and episdemic uncertainties are calculated in a label-wise manner and global measures are summed over label-wise uncertainties. Main contributions are summarized in Theorems 3.1 and 3.2, showing that the label-wise entropy-based measures satisfy desirable properties but A5 and the label-wise variance-based measures satisfy all of A0-A7.

**Q2-3 Extent To Which Claims Are Supported By Evidence:**

2: Fair: the main claims are somewhat supported by evidence (but the experimental evaluation may be weak, or does not match entirely with the claims, important baselines may be missing, proofs contain important ideas but lack rigor, algorithmic details are only discussed superficially, references are imprecise, assumptions are not sufficiently motivated or explicated, etc.).

**Q2-4 Reproducibility:**

2: Fair: key resources (e.g. proofs, code, data) are unavailable but key details (e.g. proof sketches, experimental setup) are sufficiently well-described for an expert to confidently reproduce the main results.

**Q3 Main Strengths:**

- Total, aleatoric, and episdemic uncertainties are well studied in label-wise entropy-based and variance-based perspectives.
- Axioms presented in earlier work are considered, showing which of them are satisfied by entropy-based or variance-based measures constructed in a label-wise manner.

**Q4 Main Weakness:**

- It is not clear how label-wise entropy-based measures are different from the global measures (3), (4), (6), since (17)-(19) are a common way of calculating them.
- Emphasizing label-wise uncertainty quantification could be misleading. I believe the main contribution of this paper is in the variance-based measures. But for variance-based measures, label-wise consideration is necessary and is already used in practice.

**Q5 Detailed Comments To The Authors:**

- Regarding the entropy-based measures described in (17)-(19), it is common to calculate global measures (3), (4), (6) in a label-wise manner such as (17)-(19), isn't it? Otherwise please elaborate any distinctiveness of (17)-(19), compared to (3), (4), (6).
- In practice, it is common to use variance-based measures to quantify episdemic uncertainties of classifiers. To this end, one must consider label-wise quantify, so emphasizing label-wise perspective could be misleading the contribution of this paper. As an example of recent variance-based measures, you can see this paper.
     Duan et al. (2024), "Evidential Uncertainty Quantification: A Variance-Based Perspective," WACV.
- Since  Theorems 3.1 and 3.2 are the main contributions, experiments should also demonstrate the behaviors of proposed measures to see what benefits are brought when axioms are satisfied by measures constructed a label-wise manner.

**Q9 Complying With Reviewing Instructions:**

Yes

---

> ### Author Rebuttal · Authors · 2024-04-04
>
> Thank you for your insightful review. In the following we address your concerns and questions.
>
> >It is not clear how label-wise entropy-based measures are different from the global measures (3), (4), (6), since (17)-(19) are a common way of calculating them.
>
> While we are still computing entropy, conditional entropy and mutual information information in the sums (17) - (19) there is a key difference:
>
> “Common” entropy-based measures (3), (4) and (6) are calculated on the basis of a given second-order distribution $Q$, whereas global uncertainty measures (17) - (19) derived from a label-wise construction (as we outline in more detail in Section 3) are associated with the corresponding marginal distributions $Q_k$ for $k \in {1,\dots,K}$, allowing label-wise uncertainty measures to capture uncertainty on a class level. Technically, the difference is also quite easy to see. For example, for a uniform distribution on three classes, (3) gives the entropy H(⅓, ⅓, ⅓), while our binary construction (17) yields H(⅓, ⅔) + H(⅓, ⅔) + H(⅓, ⅔), which is clearly not the same.
>
> >Emphasizing label-wise uncertainty quantification could be misleading. I believe the main contribution of this paper is in the variance-based measures. But for variance-based measures, label-wise consideration is necessary and is already used in practice.
>
> We are happy to address your concerns and clarify our position on these aspects. We agree that the theoretical and empirical study of variance-based measures is one of the contributions of our paper. However, the variance-based approach only becomes possible through the label-wise uncertainty framework, as it is a specific instantiation of the latter. As you confirm, it nicely shows that our framework enables the use of measures that cannot be directly applied to the categorical case. Moreover, please note that also show advantages of the label-wise for other measures, including variance, which can even be applied to categorical distributions. Therefore, we believe that our idea of label-wise uncertainty quantification is not only original but also important.
>
> > In practice, it is common to use variance-based measures to quantify episdemic uncertainties of classifiers. To this end, one must consider label-wise quantify, so emphasizing label-wise perspective could be misleading the contribution of this paper. As an example of recent variance-based measures, you can see this paper. Duan et al. (2024), "Evidential Uncertainty Quantification: A Variance-Based Perspective," WACV.
>
> We argue that it is indeed common to use variance-based uncertainty measures in regression settings (e.g. [1]), while it is not quite widespread in the classification setting, since variance is an example of a measure that can be applied to the binary case but not to the categorical case in general.
>
> We thank the reviewer for pointing out this recent work [2], which we will definitely incorporate in our discussion of related work.
> Still we advocate that our contribution is distinct in nature, and we do not agree that the perspective could be misleading.
> As mentioned before, we do not propose label-wise uncertainty quantification solely for the purpose of introducing variance-based measures, but offer general construction principles for such a label-wise perspective. Secondly, one other contribution is that we highlight that such measures adhere to axiomatic principles, as advocated in recent machine learning literature (see e.g. [3,4]).
>
>
> [1]  Depeweg, S., Hernandez-Lobato, J. M., Doshi-Velez, F., & Udluft, S. (2018, July). Decomposition of uncertainty in Bayesian deep learning for efficient and risk-sensitive learning. In International conference on machine learning (pp. 1184-1193). PMLR.
>
> [2]  Duan et al. (2024), "Evidential Uncertainty Quantification: A Variance-Based Perspective," WACV.
>
> [3]  Wimmer, L., Sale, Y., Hofman, P., Bischl, B., & Hüllermeier, E. (2023, July). Quantifying aleatoric and epistemic uncertainty in machine learning: Are conditional entropy and mutual information appropriate measures?. In Uncertainty in Artificial Intelligence (pp. 2282-2292). PMLR.
>
> [4]  Sale, Y., Bengs, V., Caprio, M., & Hüllermeier, E. (2023). Second-order uncertainty quantification: A distance-based approach. arXiv preprint arXiv:2312.00995.

---

### Official Review · Reviewer_TX8B · 2024-03-18

**Q2-1 Originality-Novelty:** 2
**Q2-2 Correctness-Technical Quality:** 3
**Q2-5 Clarity Of Writing:** 2

**Q1 Summary And Contributions:**

The submission addresses the problem of uncertainty quantification for classification models, in a label-wise manner. The purpose is to provide refined support to experts or decision-makers. The contributions are label-wise definitions for the three kinds of uncertainty classically encountered (total, aleatoric, epistemic), two instantiations of the proposed kinds of uncertainty (leading to entropy and variance as total uncertainty measures) as well as a short study of their properties (as per a list of axioms provided by the authors), and a series of experiments to study the properties of their proposals.

The introduction presents the setting, recalls several works related to uncertainty quantification, progressively introduces the proposal, and clearly states the problem addressed and the contributions. Section 2 presents the two-stage framework used by the authors (a first level to quantify aleatoric uncertainty, a second level for epistemic uncertainty), recalls basic knowledge about entropy-based measures, and recalls a list of axioms that should be satisfied by uncertainty measures. Section 3 proceeds with the proposal—label-wise uncertainty quantification—and studies more particularly two instantiations, which lead to entropy and variance as total uncertainty measures, which are then studied. Section 4 reports experiments on real data: first, the proposed label-wise quantification approach is evaluated on a dataset of PET/CT images; then, both proposals are compared on the CIFAR-10 dataset; eventually, their application to OOD detection is studied. Section 5 concludes the paper.

**Q2-3 Extent To Which Claims Are Supported By Evidence:**

2: Fair: the main claims are somewhat supported by evidence (but the experimental evaluation may be weak, or does not match entirely with the claims, important baselines may be missing, proofs contain important ideas but lack rigor, algorithmic details are only discussed superficially, references are imprecise, assumptions are not sufficiently motivated or explicated, etc.).

**Q2-4 Reproducibility:**

3: Good: key resources (e.g. proofs, code, data) are available and key details (e.g. proofs, experimental setup) are sufficiently well-described for competent researchers to confidently reproduce the main results.

**Q3 Main Strengths:**

The paper addresses a very interesting topic, namely (aleatoric and epistemic) uncertainty quantification. To my knowledge, quantifying uncertainty in a label-wise manner is new.

The experimental part is interesting, and sheds some light on the proposal.

**Q4 Main Weakness:**

In my opinion, several key elements in the proposal lack justification. In particular, the label-wise uncertainty measures defined by Equations (9) to (11) are not theoretically grounded, and may be questioned (see below).

The paper is not as clear as one would expect. The writing should be improved, and especially notations.

The experiments do not feature any comparison to some baseline; whereas the proposal may be original (and arguably the first attempt to address label-wise uncertainty quantification), it may have been compared to a 'naive' approach (for instance, with uncertainty being quantified on class-conditional distributions).

**Q5 Detailed Comments To The Authors:**

As mentioned above, the definitions for TU, AU, and EU (as per Eqs. (9) to (11), for instance) lack a theoretical justification. Can you provide any arguments, principles, rationales leading to these definitions ? In particular, it seems that Eq. (9) corresponds to a somewhat optimistic view: the uncertainty corresponding to the most favorable prediction is retained. Can you provide insights regarding this choice ?

As well, the relationship between $EU(Q_k)$ and $AU(Q_k)$ seems questionable; more precisely, it seems strange that both add up to TU, since they correspond to two different levels in the "uncertainty process". Can you provide comments regarding this choice ?

The log-loss being linear, and the squared-error loss quadratic, in $y\in\mathcal{Y}$, computing the expectations goes smoothly and leads to the (meaningful) results presented in the paper. However, we might expect that things become more complicated when other loss functions are used. Can you provide insights regarding how the results proposed in the paper can be extended to other settings ?

The writing may be substantially improved. Some notations are not formally defined (such as, e.g., $\Omega$, although the meaning can be easily inferred). The notations are very confusing; for instance, $\boldsymbol{\theta}$ and $\boldsymbol{\Theta}$ are used throughout Section 3 (page 5), with the notations $\theta_k$ and $\Theta_k$ being seemingly used indifferently. There are repetitions in the text, with notions being re-defined (like the expectation $\overline{\theta}_k$ page 5, whereas $\overline{\theta}$ had already been page 3). Overall, the english is good, although several typos can be found (e.g., capitals in a sentence: "[...] following template: First", or repetitions: "One way to define these measures in a meaningful way").

**Q9 Complying With Reviewing Instructions:**

Yes

---

> ### Author Rebuttal · Authors · 2024-04-03
>
> Thank you for carefully reviewing our paper. We address your comments and questions in the following.
>
> >The experiments do not feature any comparison to some baseline [..]
>
> First of all thank you for pointing out the novelty of our proposal.
>
> You rightly point out that our proposal is the first attempt to address label-wise uncertainty. We therefore only compared against the multi-class versions (if applicable, like for entropy), which we consider as natural baselines.
>
> We are not sure we completely understand what you are referring to as a *naive* approach. If you could elaborate on this, we would be happy to include it in the revised version of the paper.
>
> >As mentioned above, the definitions for TU, AU, and EU (as per Eqs. (9) to (11), for instance) lack a theoretical justification [..]
>
> Good point. We agree that this should be made a bit clearer in the paper, and that we should elaborate a bit more on the underlying intuition and rationale.
>
> Our construction can be seen as a generalization of the well-established information theoretic decomposition of Shannon entropy (TU) into conditional entropy (AU) and mutual information (EU), which are by now the most widely used measures for uncertainty quantification in machine learning. They are obtained as a special case if our loss $\phi$ is the log-loss. Our generalization consists of allowing losses other than log-loss, such as variance. Please note that we also show desirable properties in Theorems 3.1 and 3.2.
>
> >In particular, it seems that Eq. (9) corresponds to a somewhat optimistic view: the uncertainty corresponding to the most favorable prediction is retained. Can you provide insights regarding this choice?
>
> This optimism is inherent in the idea of quantifying uncertainty in terms of “unavoidable loss”. Consider the following game: You are given a (second-order) distribution $Q_k$, and you know that a distribution $\theta_k$ will be sampled from that distribution. You have to guess a $\hat{\theta}$ and will then be penalized by the loss $\phi(\hat{\theta}, \theta_k)$. Of course, you try the best you can to keep the expected loss low, hence the min in (9). If you manage to do so, this means that $Q_k$ must be “peaked” (close to deterministic), and the uncertainty is small. Otherwise, if $Q_k$ is widely spread and not very informative, the uncertainty is high, and you cannot guarantee a low loss, even with the best prediction $\hat{\theta}$. A corresponding paragraph will be added to the final version.
>
> >As well, the relationship between $EU(Q_k)$ and $AU(Q_k)$ seems questionable [..]
>
> This is again a good point, and indeed, the additive relationship between the losses has been discussed controversially in the literature (Wimmer 2023, UAI). Nevertheless, it can be justified as follows: Let TU be defined as the unavoidable loss in the prediction of $\theta_k$ as defined above. This loss comprises an epistemic component: the true data-generating process $\theta_k$ is not known and only characterized by $Q_k$. This epistemic uncertainty would disappear if $\theta_k$ were revealed. In that case, only aleatoric uncertainty would remain: If $\theta_k$ is known, the best prediction is $\hat{\theta} = \theta_k$, at least for scoring rules $\phi$, so the loss is $\phi(\theta_k , \theta_k)$; for log-loss, for example, this gives Shannon entropy. It is therefore sensible to define AU as the expectation (with regard to $Q_k$) of this remaining loss. This is Equation 10. Then, EU is naturally measured by the difference between the two: It is telling us how much (in expectation) the unavoidable loss can be reduced by getting rid of epistemic uncertainty.
>
> >The log-loss being linear, and the squared-error loss quadratic, in $y \in \mathcal{Y}$ [..]
>
> We are not entirely sure to understand what you mean by “more complicated” and “other settings”. Indeed, it will clearly be interesting to instantiate our framework with other proper scoring rules as loss functions. Note however that, because Y is binary, the loss can always be rewritten as a linear function of Y. For example the square loss can be written as $(\theta - Y)^2 = (\theta - 1)^2 Y + \theta^2 (1 - Y)$. How individual axioms depend on general properties of the loss function is indeed an interesting question, but goes beyond the scope of the current paper.
>
> >The writing may be substantially improved [..]
>
> There seems to be a misunderstanding: Please note that we are explicitly *not* using e.g. $\theta$ and $\theta_k$ interchangeably, the same goes for $\Theta$ and $\Theta_k$. While $\Theta$ refers to a random first-order distribution, distributed according to a second order distribution $Q$, by $\Theta_k$ we denote the random variable which is distributed according to $Q_k$, the marginal distribution of $Q$. We will make this distinction more prominent in the revised version.
>
> We hope that our answer clarifies your concerns about the notation and presentation.

---

### Official Review · Reviewer_YqEE · 2024-03-22

**Q2-1 Originality-Novelty:** 3
**Q2-2 Correctness-Technical Quality:** 2
**Q2-5 Clarity Of Writing:** 3

**Q1 Summary And Contributions:**

This manuscript addresses uncertainty quantification, decomposing total uncertainty into aleatoric  and epistemic uncertaincy separately for each class. A concrete motivation for doing so goes along the following lines: If there is a high epistemic uncertainty for a particular class, then gathering (or annotating) more training data specifically for that class should be prioritized.
The manuscript states 5 axioms from literature that uncertainty measures should fulfill, adds a sixth and seventh axiom, and proves that the new label-wise uncertainty measures do fulfill said axioms. Furthermore, it showcases a practical application in the context of CT image data classification.

**Q2-3 Extent To Which Claims Are Supported By Evidence:**

3: Good: the main claims are supported by convincing evidence (in the form of adequate experimental evaluation, proofs, (pseudo-)code, references, assumptions).

**Q2-4 Reproducibility:**

3: Good: key resources (e.g. proofs, code, data) are available and key details (e.g. proofs, experimental setup) are sufficiently well-described for competent researchers to confidently reproduce the main results.

**Q3 Main Strengths:**

- Topic-wise it' a perfect fit for UAI.
- Looking at the uncertainty in a label-wise manner could indeed have some benefits.
- The paper is a nice mix of theory and experiments.
- It includes an application on medical data as opposed to relying only on MNIST
- Code is available

**Q4 Main Weakness:**

- Overall, it would be nice to demonstrate more concrete use-cases for the having label-wise uncertainties beyond the gatering/annotating more data argument. The intro talks a bit about this, but it's very brief.
- The empirical studies in Section 4.1 are interesting, but a bit brief. I would prefer this part to be extended, even at the expense of 4.2 and 4.3.
- It looks like the classes are treated independently, e.g., the multiclass prediction is broken down in several binary class problems. It's not quite obvious that this is valid, given that in a multiclass setting the predictions are mutually exclusive. Hence, wouldn't we expect the uncertainties also to be dependent, e.g. there cannot be just one class with high uncertainty?
- Writing and presentation could be generally improved.

**Q5 Detailed Comments To The Authors:**

Regarding the empirical studies:
- Question: Regarding the shown CT scan images: What are the ground/truth classes, what are the predictions by the model?
- The paper currently showcases an example where the uncertainty quantification for a single patient (high EU for Lung cancer and Lymphoma) suggests the decision-making regarding gathering/annotating data (more for these two classes). But if we look at Fig. 4 (Appendix C1), we observe that there is also a patient with a high epistemic uncertainty for Melanoma. So it seems to me that, when refining training data, these decisions should be made based on all cases the model is applied on, and classes should be preferred if they have a high EU on average.
- Expanding on the previous point: I'm not sure if I see a good reason to look at a single label-wise uncertainty plot (yet).
- Does gathering/annotating more data based on EU even help in practice, and if yes, how much? One could investigate that as well. Just withhold a certain amount of data for the initial model training, do the uncertainty quantification, pick the class that seems to need more data most, add it to the data set, and show that EU is reduced for that class (and not substantially increased for others as a compensating effect).
- How does the label-wise EU correlate with the number of training examples per class? One would naturally expect that rare classes have a higher EU.

Presentation:
- Fig. 1 is not overly helpful where it is currently located as it is introduced before even the terms epistemic and aleatoric uncertainty are defined, let alone their label-wise definitions proposed in the paper. It only makes sense for the reader in retrospect.
- The definition of the "global" measures (17)-(19) and (20)-(22) are not needed, as they directly follow from (12)-(14) and the label-wise definitions above.
- The frequent use of bullet points etc. takes up additional space for little benefit.

**Q9 Complying With Reviewing Instructions:**

Yes

---

> ### Author Rebuttal · Authors · 2024-04-04
>
> Thank you for highlighting the strengths of our paper as well as your constructive criticism.
>
> >Overall, it would be nice to demonstrate more concrete use-cases for the having label-wise uncertainties beyond the gatering/annotating more data argument. The intro talks a bit about this, but it's very brief.
>
> While we agree with the reviewer that more concrete use-cases in favor of the label-wise uncertainty quantification might indeed be quite useful, we see our work as an initial step to this endeavor by contributing theoretically as well as empirically to this subject. To the best of our knowledge, our generic constructions of label-wise uncertainty measures were not addressed so far in the literature.
> In general, whenever uncertainty associated with a specific class is of importance, we see beneficial use-cases of label-wise uncertainty quantification as for example in medical diagnoses, financial risk-management and similar safety-critical scenarios.
>
> >It looks like the classes are treated independently, e.g., the multiclass prediction is broken down in several binary class problems. It's not quite obvious that this is valid, given that in a multiclass setting the predictions are mutually exclusive. Hence, wouldn't we expect the uncertainties also to be dependent, e.g. there cannot be just one class with high uncertainty?
>
> You are right, label-wise uncertainties are not independent. But we also don’t treat them as independent. Instead, their dependence is (implicitly) taken into account, because they are derived from marginalizations of the same, global second-order distribution $Q$. This is a good point, which we will highlight in the final version of the paper. Indeed, it cannot happen then that only a single class is uncertain and not (at least) a second one. It can happen, however, that uncertainty only concerns two classes and not the others. In contrast to a global measure, our label-wise approach can nicely capture this.
>
> >Question: Regarding the shown CT scan images: What are the ground/truth classes, what are the predictions by the model?”
>
> In Figure 2 (p. 7 of our manuscript) the (hard-label) prediction, as well as the ground-truth class is lung-cancer. The probabilistic prediction is given by $(p(\text{Lung Cancer}), p(\text{Lymphoma}), p(\text{Melanoma}), p(\text{Negative})) =  (0.74, 0.24, 0.02, 0)$. In the final version of our paper, we will incorporate this information in our visualizations.
>
> >The paper currently showcases an example where the uncertainty quantification for a single patient (high EU for Lung cancer and Lymphoma) suggests the decision-making regarding gathering/annotating data (more for these two classes). But if we look at Fig. 4 (Appendix C1), we observe that there is also a patient with a high epistemic uncertainty for Melanoma [..]
>
> This is a good observation. We agree that to refine training data it might be indeed beneficial to look at label-wise EU aggregated over the data.
>
> >Expanding on the previous point: I'm not sure if I see a good reason to look at a single label-wise uncertainty plot (yet).”
>
> To give an example of the usefulness of label-wise uncertainties, we can refer again to the medical setting. In this context, it could be more important to give accurate predictions for certain classes. For example, a medical expert may only want to predict the negative class when the uncertainty with respect to this class is very low, as giving a false negative diagnosis can be detrimental. Hence, this would be a situation where it is useful to look at a single label-wise uncertainty plot.
>
> >Does gathering/annotating more data based on EU even help in practice, and if yes, how much? One could investigate that as well. Just withhold a certain amount of data for the initial model training, do the uncertainty quantification, pick the class that seems to need more data most, add it to the data set, and show that EU is reduced for that class (and not substantially increased for others as a compensating effect).
>
> That’s an interesting proposal. We will prepare and run experiments of that kind, and will hopefully be able to show some results before the end of the rebuttal period. However, since the experiments are computationally demanding, and the period is short, we can’t promise that this will work out.
>
> >How does the label-wise EU correlate with the number of training examples per class? One would naturally expect that rare classes have a higher EU.
>
> We agree with your observation. From the conceptual understanding that epistemic uncertainty is directly tied to the model's lack of knowledge about the data-generating process, and generally can be reduced in terms of additional information (viz. more data), we can expect that rare classes will have higher EU.
>
>
> We further thank the reviewer for his suggestions in terms of presentations. We will make sure to take them into account in the final version.

---

### Official Review · Reviewer_uxTi · 2024-03-22

**Q2-1 Originality-Novelty:** 3
**Q2-2 Correctness-Technical Quality:** 3
**Q2-5 Clarity Of Writing:** 4

**Q1 Summary And Contributions:**

This paper considers uncertainty quantification in the context of classification. In a Bayesian framework, probabilistic classification starts with a posterior distribution over the possible classes. This captures aleatoric uncertainty, or uncertainty due to the inherent randomness of the data generating process. However because this is a single distribution, it does not capture epistemic uncertainty, or uncertainty due to finite sample size issues.

The authors provide an advance upon the state-of-the-art by providing a label-wise decomposition of this perspective. This has certain methodological advantages (allows different loss functions). A good amount of experiments are provided to demonstrate the utility of this work.

**Q2-3 Extent To Which Claims Are Supported By Evidence:**

4: Excellent: all claims are supported by very convincing evidence (in the form of comprehensive experimental evaluation, rigorous mathematical proofs, detailed (pseudo-)code, precise references, well-motivated and realistic assumptions) and the authors deliver what they promise.

**Q2-4 Reproducibility:**

4: Excellent: key resources (e.g. proofs, code, data) are available and key details (e.g. proof sketches, experimental setup) are comprehensively described for competent researchers to confidently and easily reproduce the main results.

**Q3 Main Strengths:**

This is a clearly written paper (even to a relative outsider) that explores a problem that I expect to be of interest to a substantial community - higher-order quantification of uncertainty. The advancement is substantive and novel, and there are substantial experiments to back up the paper. I expect interested researchers would have no problem reproducing this paper given the code and appendices.

**Q4 Main Weakness:**

I don't have any major qualms about this paper.

**Q5 Detailed Comments To The Authors:**

No major criticisms of what appears to be a well-written paper on an interesting topic.

Minor typo on page 3: KL divergences are defined by are not otherwise mentioned?

**Q9 Complying With Reviewing Instructions:**

Yes

---

> ### Author Rebuttal · Authors · 2024-04-04
>
> First of all, thank you very much for appreciating the quality and strengths of our paper.
>
> >Minor typo on page 3: KL divergences are defined by are not otherwise mentioned?
>
> While we need KL divergences later on (for entropy-based uncertainty measures), you rightfully point out that this definition is misplaced in the text. Thank you for pointing out, we will streamline accordingly.

---

### Meta-Review · Area_Chair_q11n · 2024-04-17

This paper introduces a method for label-wise uncertainty quantification for classification tasks.  The method quantifies both the aleatoric and epistemic uncertainties for each class.  The reviewers like this work; the paper receives 5 reviews, including 2 strong accepts, 2 weak accepts, and 1 borderline accept.  The paper's main strength lies in its calculating label-wise uncertainties instead of the total uncertainties over all classes, a significant advance over the existing work.  Other strengths include theoretical proof of the proposed method and substantial experiments that demonstrate the effectiveness of the proposed method.  Weaknesses of this work include the unrealistic class independent assumption, unclear writing about some aspects of the proposed method, and missing comparison with some baseline methods.  The author’s rebuttal is detailed and effective.  It addresses most of these concerns.